# Heterogeneity in VEGFR3 levels drives lymphatic vessel hyperplasia through cell-autonomous and non-cell-autonomous mechanisms

Yan Zhang[1], Maria H. Ulvmar [1], Lukas Stanczuk[1], Ines Martinez-Corral [1], Maike Frye[1], Kari Alitalo[2] & Taija Mäkinen[1]

Incomplete delivery to the target cells is an obstacle for successful gene therapy approaches. Here we show unexpected effects of incomplete targeting, by demonstrating how heterogeneous inhibition of a growth promoting signaling pathway promotes tissue hyperplasia. We studied the function of the lymphangiogenic VEGFR3 receptor during embryonic and post-natal development. Inducible genetic deletion of *Vegfr3* in lymphatic endothelial cells (LECs) leads to selection of non-targeted VEGFR3+ cells at vessel tips, indicating an indispensable cell-autonomous function in migrating tip cells. Although *Vegfr3* deletion results in lymphatic hypoplasia in mouse embryos, incomplete deletion during post-natal development instead causes excessive lymphangiogenesis. Analysis of mosaically targeted endothelium shows that VEGFR3− LECs non-cell-autonomously drive abnormal vessel anastomosis and hyperplasia by inducing proliferation of non-targeted VEGFR3+ LECs through cell-contact-dependent reduction of Notch signaling. Heterogeneity in VEGFR3 levels thus drives vessel hyperplasia, which has implications for the understanding of mechanisms of developmental and pathological tissue growth.

---

[1] Department of Immunology, Genetics and Pathology, Uppsala University, Dag Hammarskjölds väg 20, 751 85 Uppsala, Sweden. [2] Wihuri Research Institute and Translational Cancer Biology Program, Biomedicum Helsinki, University of Helsinki, FIN-00014 Helsinki, Finland. These authors contributed equally: Yan Zhang, Maria H. Ulvmar.  Correspondence and requests for materials should be addressed to T.Mäk. (email: taija.makinen@igp.uu.se)

**B**lood and lymphatic vessels play essential roles during development and adult tissue homeostasis, as well as in various diseases[1–3]. During development and vascular growth, most blood and lymphatic vessels form by sprouting from pre-existing vessels through a process termed (lymph) angiogenesis. Previous studies have established a critical role for Notch signaling during sprouting angiogenesis in the specification of endothelial cells (ECs) to migratory tip cells and proliferating stalk cells that show differential sensitivities and

responses to angiogenic growth factors[4]. Blood endothelial tip cells express high levels of the vascular endothelial growth factor receptor 2 (VEGFR2) and VEGFR3, and upregulate the Notch ligand delta-like 4 (DLL4)[5–8]. DLL4 has, in turn, been thought to induce stalk cell behavior in adjacent ECs by activating Notch signaling[9–11]. Recent studies however indicate that activation of Notch signaling is required in tip cells, and the level of Notch activity is more important than direct DLL4-mediated cell–cell communication in controlling EC behavior during sprouting

**Fig. 1** VEGFR3 is indispensable for tip cell function during embryonic dermal lymphatic vessel sprouting. **a** Schematic of the genetic constructs and 4-OHT administration schedule (Cre induction, red (6 × 1 mg); estimated effective period (24 h), light red). The timing of dermal lymphatic vessel formation in the dorsal skin is indicated. **b** Whole-mount immunofluorescence of E17.5 *Vegfr3^flox;R26-mTmG;Prox1-CreER^T2* skin. Boxed areas are magnified and single channel images for VEGFR3 staining are shown. Note efficient depletion of VEGFR3 in the Cre-targeted (GFP+) LECs and the presence of non-targeted cells at the tips of hypoplastic vessel sprouts (arrows) in the mutant (*flox/flox*) skin. **c–g** Quantification of dermal lymphatic vessel parameters in E17.5 *Vegfr3^flox;R26-mTmG;Prox1-CreER^T2* embryos. Bars represent mean (*n* = 4–10 embryos, as indicated) ± s.e.m. **h**, **i** Whole-mount analysis of E16.5 skin after mosaic deletion of one (*flox/+*) or two (*flox/flox*) *Vegfr3* alleles by a single 4-OHT (1 mg) administration at E13.5. **j** Quantification of Cre-targeted (GFP+) LECs at the dorsal midline (area depicted in **i**). Bars represent mean (*n* = 5–6 embryos, as indicated) ± s.e.m. * *P* < 0.05, *** *P* < 0.001. Two-tailed unpaired Student's *t* test (**c–g**, **j**). Scale bars: 200 μm (**b**, **i**). ns: not significant

angiogenesis[12,13]. Nevertheless, inhibition of DLL4-Notch signaling leads to vascular hypersprouting in vivo[9–11]. Abnormal vascular responses leading to excessive sprouting are often associated with increased EC proliferation. How cell migration and proliferation are coupled during vascular network formation is however incompletely understood.

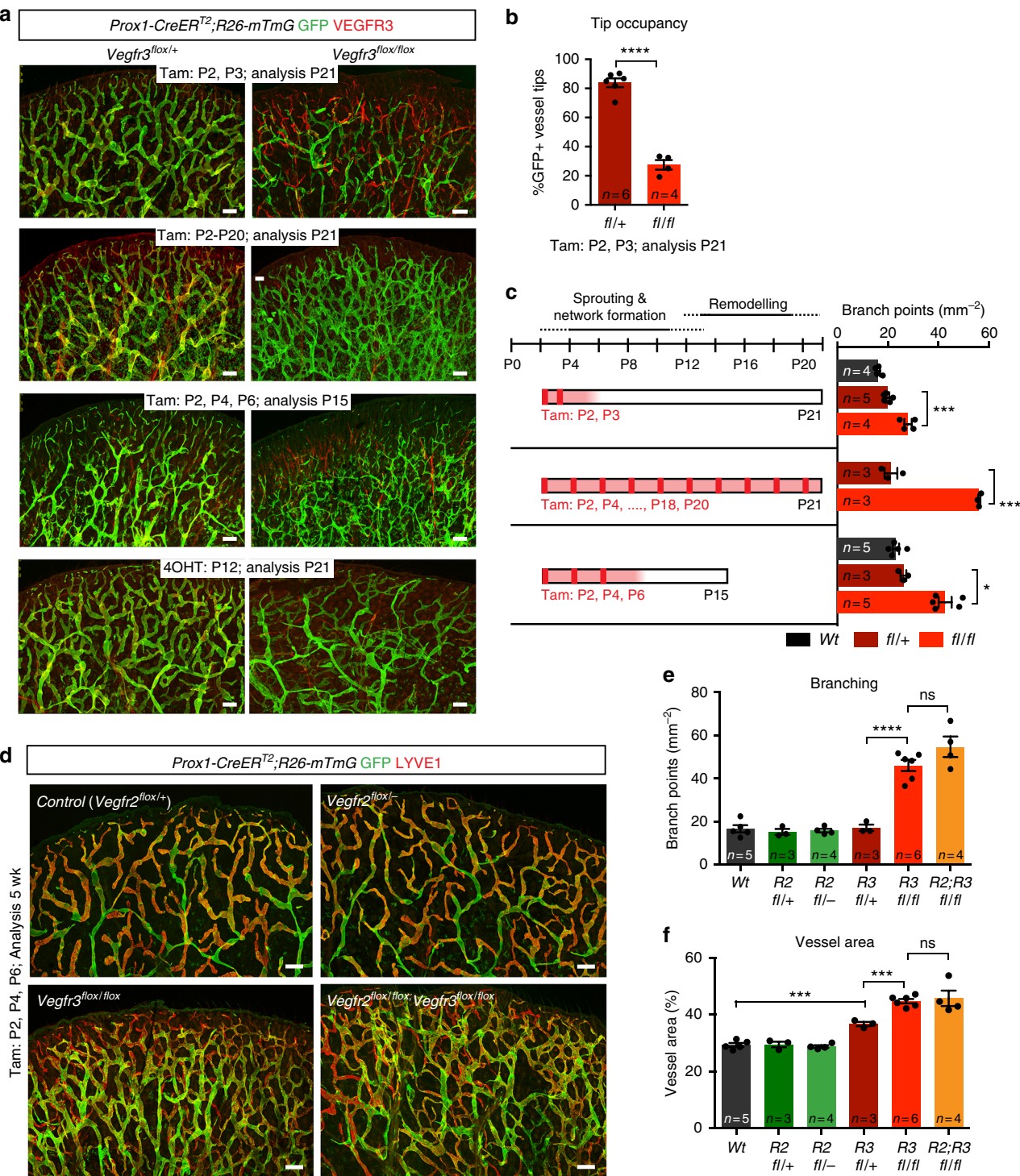

**Fig. 2** Early post-natal deletion of *Vegfr3* results in dermal lymphatic vessel hyperplasia during a critical post-natal period of 2 weeks. **a** Time course analysis of the effect of post-natal *Vegfr3* deletion on dermal lymphatic development. Whole-mount immunofluorescence of ear skin of indicated stages and Tam/4-OHT treatment regimes. **b** Quantification of Cre-targeted (GFP+, *Vegfr3* deleted) LECs in the distal vessel tips in P21 mice treated with Tamoxifen at P2 and P3. Bars represent mean ($n = 6$ *fl/+* and $n = 4$ *fl/fl* mice) ± s.e.m. **c** Quantification of vessel branch points in ear skin analyzed in Fig. 2a after indicated Tam treatment regimes. Tamoxifen administration schedule (Cre induction, red ($n \times 150$ μg); estimated effective period (72 h), light red) (left) is shown. Bars represent mean ($n = 3$–5 mice, as indicated) ± s.e.m. **d** Whole-mount immunofluorescence of ear skin of 5 weeks old mice treated with Tamoxifen at P2, P4 and P6. **e, f** Quantification of vessel branch points and area in ear skin analyzed in Fig. 2d. Bars represent mean ($n = 3$–6 mice, as indicated) ± s.e.m. *$P < 0.05$, ***$P < 0.001$, ****$P < 0.0001$. Two-tailed unpaired Student's *t* test (**b, c, e, f**). Scale bars: 200 μm (**a, d**). ns: not significant

The key regulator of lymphatic vascular growth and lymphangiogenic vessel sprouting is VEGF-C[14–17], but the relative contributions of its two tyrosine kinase receptors VEGFR2 and VEGFR3 have not been investigated in detail. Here we studied the

function of VEGFR3 during embryonic and post-natal lymphangiogenesis using conditional Cre/loxP mediated gene deletion in mice. We found that VEGFR3 is required cell-autonomously for lymphatic endothelial tip cell function and

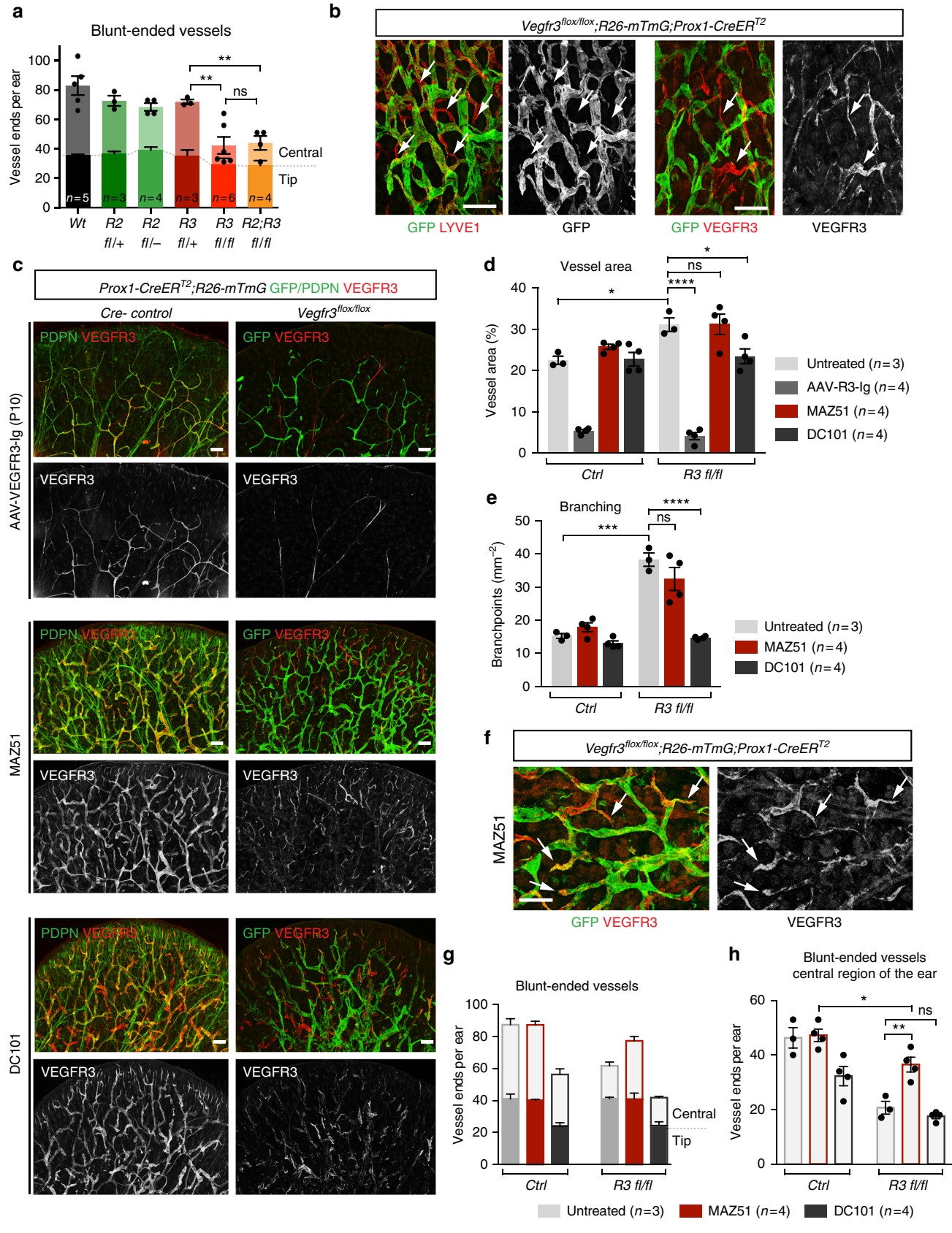

vessel sprouting. Unexpectedly, VEGFR3 downregulation is coupled to lateral induction of cell proliferation and vessel anastomosis through inhibition of Notch signaling in the neighboring VEGFR3+ LECs that escaped Cre recombination. These data uncover a previously unrecognized non-cell-autonomous mechanism regulating vascular growth.

## Results

**VEGFR3 is key to lymphatic endothelial tip cell function**. To investigate the cellular mechanisms of lymphatic vessel sprouting, we genetically deleted *Vegfr3* in lymphatic endothelia by crossing mice carrying conditional *Vegfr3flox* alleles with *Prox1-CreERT2* animals (Fig. 1a). Gene-deleted cells were visualized by simultaneous Cre-mediated activation of the green fluorescent protein (GFP) using the *R26-mTmG* reporter line. We administered 4-OHT on six consecutive days starting at E10.5 to ensure efficient gene targeting (Fig. 1a), which was evidenced by uniform GFP expression in the lymphatic vessels of control embryos at E17.5 (Fig. 1b). In *Vegfr3flox/flox;R26-mTmG;Prox1-CreERT2* embryos, GFP expression coincided with efficient depletion of VEGFR3 protein (Fig. 1b, Supplementary Fig. 1), and reduced vessel branching and diameter (Fig. 1b–d). Analysis of the sprouting front at the dorsal midline of the skin revealed that most lymphatic vessel tips were occupied by GFP+ (i.e., gene targeted) LECs in the control embryos. In contrast, the majority of *Vegfr3flox/flox* vessels had a non-recombined VEGFR3+ cell at the tip (Fig. 1b, e). The rare vessel tips composed of *Vegfr3*-deleted cells showed a "blunted" morphology, in contrast to a "spiky" morphology observed in most vessel tips in E17.5 control skin (Fig. 1f). Consistent with the reduced ability of *Vegfr3*-deficient cells to contribute to vessel tips, the extension of lymphatic vessels to the dorsal midline was compromised (Fig. 1g).

Next, we assessed the competence of LECs with homozygous or heterozygous deletion of *Vegfr3* to compete with wild-type neighboring cells during embryonic lymphatic vessel sprouting. Mosaic deletion of either one or two alleles of *Vegfr3* was induced by a single dose of 4-OHT at E13.5 (Fig. 1h) to label LECs preferentially in the lateral skin[18]. The ability of gene targeted GFP+ cells to migrate in the distally extending sprouts was assessed at E16.5 (Fig. 1i). Both heterozygous and homozygous *Vegfr3* null cells failed to migrate within the sprouts to the dorsal midline area (Fig. 1i, j). Together, the above results demonstrate that VEGFR3 is indispensable for tip cell function during embryonic dermal lymphatic vessel sprouting.

**Early post-natal *Vegfr3* deletion causes lymphatic hyperplasia**. To investigate if the mechanisms of lymphatic vessel sprouting are conserved across stages of development, we studied the dependence of early post-natal lymphatic development on VEGFR3. We analyzed the lymphatic vasculature on the dorsal side of the ear that forms by sprouting after post-natal day (P)4 and is remodeled from a primary vascular plexus into a network of blind-ended lymphatic capillaries and valve-containing

collecting vessels between P12–P21[19,20]. *Vegfr3* deletion was induced prior to initiation of lymphatic sprouting by two consecutive administrations of Tamoxifen at P2 and P3 (Fig. 2a). We considered that serum levels of the active metabolite 4-OHT are maintained up to 72 h after each administration[18,21], i.e., until P6. GFP expression, correlating with VEGFR3 depletion in the mutant mice (Supplementary Fig. 2), revealed an incomplete contribution of the *Vegfr3*-deleted cells to the lymphatic vasculature and their exclusion from the distal tips of lymphatic capillaries (Fig. 2a, b). Surprisingly, a modest increase in vessel hyperbranching was observed in the mutants (Fig. 2c). To maximize gene targeting efficiency, we administered Tamoxifen every second day until P21. This unexpectedly resulted in a severely hyperbranched lymphatic vessel network (Fig. 2a, c), a phenotype opposite to that observed upon embryonic loss of *Vegfr3* (Fig. 1a–g) or systemic inhibition of VEGFR3 or its ligands[22]. Additional temporally controlled gene deletions showed that three Tamoxifen administrations at P2, P4, and P6 were sufficient to induce efficient gene targeting and vessel hyperbranching, which was evident already at P15 (Fig. 2a, c). As expected, and confirming the LEC-autonomous nature of the phenotype, dermal blood vasculature was not affected in the mutants (Supplementary Fig. 3a). A similar lymphatic vessel hyperbranching phenotype was observed in the intestinal wall of mutant mice (Supplementary Fig. 3b, c). We conclude that early post-natal *Vegfr3* deletion leads to lymphatic vessel hyperplasia during the formation of the primary dermal vascular plexus in a critical post-natal period of 2 weeks. This was further supported by the finding that a single 4-OHT administration at P12, when the remodeling of the dermal lymphatic vasculature of the ear begins[19], did not induce vessel hyperplasia despite efficient VEGFR3 depletion (Fig. 2a, Supplementary Fig. 2). Instead, a distinct phenotype characterized by abnormal patterning of the distal lymphatic capillary network was observed (Fig. 2a).

The excessive dermal lymphangiogenic growth and branching observed upon early post-natal deletion of *Vegfr3* closely resembled the angiogenic hypersprouting phenotype in the retinal vasculature resulting from endothelial *Vegfr3* deletion[7]. Blood vessel hyperplasia was linked to increased VEGFR2 protein expression and signaling, thus implicating VEGFR3 as a negative regulator of the VEGF-VEGFR2 pathway[7,23]. To test if increased VEGFR2 signaling also contributes to *Vegfr3* loss-induced lymphatic vessel hyperplasia, we genetically deleted *Vegfr2* in combination with *Vegfr3* using the *Prox1-CreERT2*. Loss of *Vegfr2* did not cause an apparent phenotype (Fig. 2d–f, Supplementary Fig. 4a–c). In addition, *Vegfr2* deletion, using a 3-day (Fig. 2d–f) or an extended 10-day Tamoxifen administration protocol (Supplementary Fig. 4a–c), did not rescue, but rather exacerbated the *Vegfr3* mutant phenotype. These results suggest that in the lymphatic vasculature, cell-autonomous regulation of VEGFR2 by VEGFR3 does not explain vessel hyperplasia induced by the loss of *Vegfr3*. This was further supported by unchanged VEGFR2 protein and messenger RNA (mRNA) expression in *Vegfr3*-deleted LECs isolated from the mutant ears (Supplementary

**Fig. 3** Lymphatic vascular hyperplasia in *Vegfr3*-deleted ears is driven by VEGF-C signaling. **a** Quantification of blunt-ended vessels (lymphatic capillaries) in 5 weeks old mice treated with Tamoxifen at P2, P4, and P6. Central region excludes 560 μm area (tip) from the edge of the ear. Bars represent mean ($n$ = 3–6 mice, as indicated) ± s.e.m. **b** Whole-mount immunofluorescence of *Vegfr3flox/flox;R26-mTmG;Prox1-CreERT2* ear skin showing vessel interconnections (arrows) formed of non-targeted (GFP−) LYVE1+ (left) and VEGFR3+ (right) LECs. **c** Whole-mount immunofluorescence of ear skin of 3 weeks old *Vegfr3flox/flox;R26-mTmG;Prox1-CreERT2* and Cre-negative littermate mice treated with Tamoxifen at P2, P4, and P6, and inhibitors of VEGF-C signaling (the VEGF-C-trap VEGFR3-Ig, the VEGFR3 inhibitor MAZ51, or the VEGFR2 blocking antibody DC101). Quantification of (**d**) vessel area and (**e**) branch points. Bars represent mean ($n$ = 3 (untreated groups) or $n$ = 4 (treated groups) mice) ± s.e.m. **f–h** Visualization by whole-mount immunofluorescence (arrows in **f**) and quantification of blunt-ended vessels in the (**g**) entire and (**h**) central region of the ear in *Vegfr3flox/flox;R26-mTmG;Prox1-CreERT2* mice treated with the VEGFR3 kinase inhibitor MAZ51. In **g**, **h**, bars represent mean ($n$ = 3 (untreated) or $n$ = 4 (MAZ51, DC101) mice) ± s.e.m. *$P < 0.05$, **$P < 0.01$, ***$P < 0.001$, ****$P < 0.0001$. Two-tailed unpaired Student's $t$ test (**a**, **d**, **e**, **h**). Scale bars: 200 μm (**b**, **c**, **f**). ns: not significant

Fig. 4d, e, Supplementary Fig. 5) and in *Vegfr3*-deficient vessels in vivo (Supplementary Fig. 4f).

**VEGF-C drives lymphatic hyperplasia in *Vegfr3*-deleted ears**. Detailed characterization of the lymphatic phenotype in the *Vegfr3^flox^;R26-mTmG;Prox1-CreER^T2^* ears revealed that the number of blunt-ended lymphatic capillaries was significantly reduced in the central region of *Vegfr3*-deleted ears in comparison to control ears (Fig. 3a), suggesting that abnormal anastomosis of lymphatic capillaries contributes to the hyperbranching phenotype. The phenotype was not altered when *Vegfr2* was also

deleted (Fig. 3a). Notably, abnormal interconnections were predominantly formed by non-targeted LECs when *Vegfr3* was deleted alone (Fig. 3b) or in combination with *Vegfr2* (Supplementary Fig. 4g). This suggested that rather than acting in a cell-autonomous manner, *Vegfr3* deficiency induces abnormal behavior of neighboring VEGFR3+ cells.

To assess the contribution of VEGF-C signaling to the phenotype, due to its ability to stimulate non-targeted LECs, we inhibited all VEGF-C signaling in the *Vegfr3^flox/flox^;R26-mTmG; Prox1-CreER^T2^* mutant and Cre⁻ littermate control mice by administering adeno-associated vectors (AAVs) encoding soluble

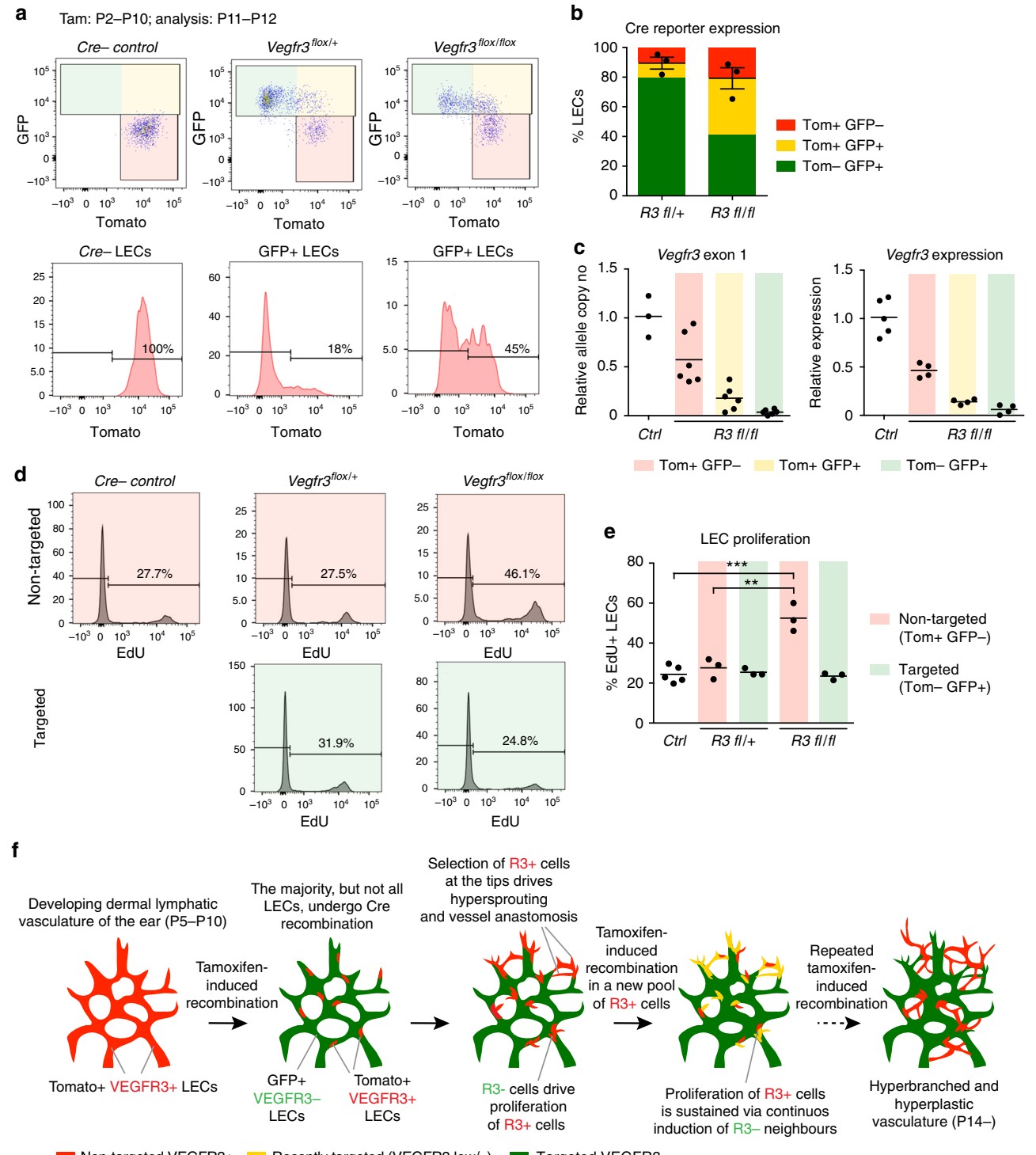

VEGF-C-trap (AAV-Vegfr3-Ig)[24]. The relative contribution of the two VEGF receptors in non-targeted LECs was further investigated by treatment with the VEGFR2 blocking antibody DC101 or the VEGFR3 kinase inhibitor MAZ51. To overcome the requirement of VEGF-C signaling for LEC survival and proliferation at early post-natal stages[22], while achieving efficient inhibition during the critical period for the phenotype to develop (Fig. 2c), we administered AAV-VEGFR3-Ig by intraperitoneal injection at P10. As expected, control mice expressing VEGFR3-Ig showed decreased growth of lymphatic vasculature in the ear skin (Fig. 3c, d). In the *Vegfr3*<sup>flox/flox</sup>;*R26-mTmG;Prox1-CreER*<sup>T2</sup> mutants, vessel hyperplasia was completely rescued and the vasculature showed a similar reduction in growth as in the control (Fig. 3c, d). Selective VEGFR3 inhibition from P7 did not affect the ear vasculature in the control mice when analyzed at P21 (Fig. 3c–e), but partially rescued the hyperbranching in 2 out of 4 mutants (Fig. 3e). Interestingly, VEGFR3 inhibition significantly reduced vessel anastomosis and restored the number of blunt-ended vessels in the central region of the ear (Fig. 3f–h). In contrast, blocking VEGFR2 signaling inhibited *Vegfr3* loss-induced hyperplasia and excessive vessel branching (Fig. 3c–e), but the number of blunt-ended vessels was not increased (Fig. 3g, h). Together, these data show that VEGF-C, through signaling via both VEGFR2 and VEGFR3 in the non-targeted LECs, drives lymphatic vascular hyperplasia in the *Vegfr3*<sup>flox/flox</sup>;*R26-mTmG; Prox1-CreER*<sup>T2</sup> mutant skin.

**Non-targeted LECs overproliferate in *Vegfr3*-deleted ears.** To further understand the cellular mechanism underlying the non-cell-autonomous effect of *Vegfr3* deletion, we focused our analysis on the critical developmental stages when the phenotype first arises. Analysis of P11–P12 *Vegfr3*<sup>flox</sup>;*R26-mTmG;Prox1-CreER*<sup>T2</sup> homozygous mutant and heterozygous control ears showed a similar overall recombination efficiency (i.e., GFP expression) in the LEC population (Fig. 4a, b). However, analysis of the proportion of LECs that had undergone a recent recombination event, as revealed by perdurance of Tomato protein in *Tomato*-gene-deleted GFP expressing cells, showed a striking difference between the genotypes. In the heterozygous controls, the majority of GFP<sup>+</sup> LECs had lost Tomato by P11–P12 when Tamoxifen was administered continuously from P2 onwards (Fig. 4a, b; indicated in green). In contrast, most GFP<sup>+</sup> LECs were Tomato<sup>+</sup> in the mutant ears (Fig. 4a, b; indicated in yellow). GFP expression and loss of Tomato was confirmed to correlate with genetic deletion of *Vegfr3* exon 1 and expression of *Vegfr3* mRNA in LECs sorted from mutant and control mice (Fig. 4c, Supplementary Fig. 5). GFP<sup>+</sup>Tomato<sup>−</sup> LECs from *Vegfr3*<sup>flox/flox</sup>;*R26-mTmG;Prox1-CreER*<sup>T2</sup> mice showed reduced exon 1 copy number (relative copy number $3.7 \pm 2.3\%$ ($n = 6$)) concomitant with reduced *Vegfr3* mRNA expression (relative expression $6.1 \pm 4.2\%$ ($n = 4$))

compared to Cre-negative controls (Fig. 4c). In contrast, GFP<sup>−</sup>Tomato<sup>+</sup> cells displayed variable levels of retained exon 1 (relative copy number $57.3 \pm 23.8\%$ ($n = 6$)) and *Vegfr3* mRNA (relative expression $46.4 \pm 6.4\%$ ($n = 4$)), suggesting loss of one *Vegfr3* allele. This was also observed in the GFP<sup>+</sup>Tomato<sup>+</sup> LEC (Fig. 4c). These data demonstrate that GFP<sup>+</sup>Tomato<sup>−</sup> LECs represent fully recombined cells that have lost *Vegfr3* expression, while GFP<sup>−</sup>Tomato<sup>+</sup> LECs retain, at the population level, *Vegfr3* expression that corresponds to, or is higher than in *Vegfr3* heterozygous LECs. The high proportion of Tomato<sup>+</sup> cells in the Cre-targeted GFP<sup>+</sup> LEC population specifically in the homozygous *Vegfr3*<sup>flox/flox</sup> mice further suggests selection against complete loss of *Vegfr3* and continuous initiation of recombination in non-targeted LECs.

To address how a pool of non-recombined LECs is maintained in the mutant skin, we analyzed LEC proliferation in the *Vegfr3*<sup>flox</sup>;*R26-mTmG;Prox1-CreER*<sup>T2</sup> ears by flow cytometry. Non-targeted LECs in the P11–P12 mutant ears showed a significantly higher EdU incorporation rate than targeted *Vegfr3*-deficient LECs, or LECs in control ears from Cre-negative littermates (Fig. 4d, e). Heterozygous or homozygous loss of *Vegfr3* did not affect LEC proliferation under these conditions (Fig. 4d, e). These results support a model (Fig. 4f) whereby incomplete Cre-mediated targeting leads to selection of residual non-targeted VEGFR3<sup>+</sup> LECs to the tips of vessel sprouts (Fig. 2b), and promotes their excessive proliferation (Fig. 4d) and vascular anastomosis (Fig. 3a, b). Proliferation of non-targeted LECs is subsequently sustained by a continuous induction of new VEGFR3<sup>−</sup> neighbors through repeated Tamoxifen-induced recombination during the critical post-natal period of 2 weeks (Fig. 4f).

**Post-natal role of VEGFR3 in LEC survival and proliferation.** Lymphatic hyperplasia caused by early post-natal *Vegfr3* deletion was unexpected considering the previously established pro-lymphangiogenic role of VEGF-C signaling during embryogenesis[14,16] (Fig. 1) and early post-natal development[17,22]. Continuous induction of *Vegfr3* deletion in a new pool of non-recombined LECs over a period of several days (Fig. 4) did not allow the follow-up of the *Vegfr3*-deleted cells and their survival. For this purpose, we performed a pulsed induction of *Vegfr3* deletion by a single dose of 4-OHT that was administered during the active lymphangiogenic growth phase at P7 or P10 (Supplementary Fig. 6a). Two days after 4-OHT injection at P7 or P10, we observed reduced proportion of GFP<sup>+</sup> LECs in *Vegfr3*-deleted compared to *Vegfr3* heterozygous skin (Supplementary Fig. 6b, c). This reduction could be referred to a selective reduction in GFP<sup>+</sup> LECs that had lost Tomato (Supplementary Fig. 6b, d, e), suggesting decreased survival of *Vegfr3*-deficient LECs (Fig. 4c) at this stage. *Vegfr3*-deficient LECs also showed significantly less

**Fig. 4** Excessive proliferation of non-targeted VEGFR3<sup>+</sup> LECs contributes to lymphatic vessel hyperplasia in *Vegfr3*-deleted ears. **a**, **b** Fluorescence-activated cell sorting (FACS) evaluation of Tomato and GFP expression in *Vegfr3*<sup>flox/+</sup> and *Vegfr3*<sup>flox/flox</sup>;*R26-mTmG;Prox1-CreER*<sup>T2</sup> LECs. Tamoxifen (150 μg) was administered daily from P2 until harvest at P11–P12. Representative dot plots of GFP and Tomato expression, and histogram of Tomato expression in GFP<sup>+</sup> LECs are displayed with Cre<sup>−</sup> littermate as a control. In **b**, graph of combined results from two litters, showing the average distribution of cells expressing Tomato only (red), GFP and Tomato (yellow), and GFP only (green). Dots show % of GFP<sup>+</sup> LECs in individual mice, horizontal line represents mean ($n = 3$ mice) ± s.e.m. **c** *Vegfr3* exon1 deletion (left) and mRNA expression (right) in sorted Tomato<sup>+</sup>GFP<sup>−</sup> (red), Tomato<sup>+</sup>GFP<sup>+</sup> (yellow), and Tomato<sup>−</sup>GFP<sup>+</sup> (green) LEC populations from P11–P14 *Vegfr3*<sup>flox/flox</sup> mutants (R3 fl/fl). Tamoxifen (150 μg) was administered at P2, P4, and P6. LECs from Cre<sup>−</sup> mice (*Ctrl*) were used as controls with the average ΔCT cycle threshold as the reference value for relative quantification. Horizontal line represents mean ($n = 3$–6 mice). **d**, **e** 5-ethynyl-2'-deoxyuridine (EdU) incorporation evaluated by FACS analysis in LECs gated on Tomato and GFP expression, shown in Fig. 3c. Representative histograms of EdU incorporation evaluated 24 h after injection and graph of combined results from two litters are shown in **e**. Horizontal line represents mean ($n = 3$–5 mice). **f** Proposed model for the mechanism underlying post-natal *Vegfr3* loss-induced lymphatic vessel hyperplasia in the ear. Incomplete deletion of *Vegfr3* promotes selection of non-targeted VEGFR3<sup>+</sup> cells to the tips of vessel sprouts and their excessive proliferation and vascular anastomosis. Proliferation of non-targeted VEGFR3<sup>+</sup> cells is sustained by continuous induction of new VEGFR3<sup>-</sup> neighbors through repeated Tamoxifen administrations during a critical post-natal period of 2 weeks. **$P < 0.01$, ***$P < 0.001$. Two-tailed unpaired Student's *t* test (**e**). ns: not significant

EdU incorporation than LECs in the control skin (Supplementary Fig. 6f), resembling the situation in E15 thoracic skin (Supplementary Fig. 6g). The requirement of VEGFR3 for LEC survival and proliferation selectively during the early post-natal development was corroborated by regression of the dermal lymphatic vasculature in the ear when VEGF-C-trap was administered at P7 (Supplementary Fig. 6h), but not when administered after P14[22]. Taken together, these results demonstrate that during the initial growth of the lymphatic vascular plexus, VEGFR3 is required for LEC survival and proliferation, both in the dorsal skin of embryos (Fig. 1) and in the ear skin of early post-natal mice (Supplementary Fig. 6).

**VEGFR3$^-$ LECs induce proliferation of VEGFR3$^+$ neighbors.**
Next, we sought to understand the mechanism by which the proliferation of non-targeted LECs is controlled in the *Vegfr3*-deleted ears. In the retina, engulfment of VEGF by VEGFR2 expressing neurons controls VEGF levels to limit blood vessel growth around neurons[25]. We asked if the inability of *Vegfr3*-deficient LECs to consume VEGF-C by receptor-mediated endocytosis and subsequent degradation leads to excessive VEGF-C levels and thereby contributes to vessel hyperplasia in the mutant ears. Assessment of VEGF-C concentrations in tissue homogenates by enzyme-linked immunosorbent assay (ELISA) did not reveal differences between control and *Vegfr3* mutant ears (Fig. 5a). *Vegfc* or the related *Vegfd* mRNA was not increased (Supplementary Fig. 7a, b), neither was the recruitment of macrophages, which are major producers of VEGF-C and VEGF-D in vivo[26,27] (Supplementary Fig. 7c).

We then asked if *Vegfr3*-deleted LECs instead induce proliferation of neighboring wild-type cells through direct cell–cell communication. We isolated primary dermal LECs from adult *Vegfr3$^{flox/flox}$;R26-mTmG;Prox1-CreER$^{T2}$* mice and analyzed cell proliferation after 4-OHT mediated *Vegfr3* deletion in vitro. GFP expression, which correlated with VEGFR3 depletion (Fig. 5b), was observed in $11 \pm 1.4$ % ($n = 6$) of LECs 3–4 days after 4-OHT treatment. Under confluent contact-inhibited conditions with low proliferation rate, most EdU$^+$ wild-type (WT) cells were in direct contact with VEGFR3-deleted (KO) cells (Fig. 5c). To monitor the acute response of LECs to VEGFR3 deletion, we mixed cell tracker-labeled human dermal LECs treated with control siRNA (siCTRL) with non-labeled LECs treated with VEGFR3 siRNA (siVEGFR3) (Fig. 5d). Increasing the proportion of siVEGFR3 cells in the co-cultures increased proliferation of the siCTRL LECs (Fig. 5e). To ask if siCTRL cells were more likely to proliferate when having immediate siVEGFR3 neighbors, we calculated the probability of random association. Under confluent culture conditions, LECs had an average of five neighbors ($5.2 \pm 0.2$, $n = 172$ LECs). In 80:20% siCTRL:siVEGFR3 co-cultures, the predicted proportion of siCTRL LECs that were in direct contact with at least one siVEGFR3 LEC was therefore 67.2%, which was in agreement with our experimental observation ($63.4 \pm 3.2\%$, $n = 5$) (Fig. 5f). However, the proportion of EdU$^+$ siCTRL cells that were in direct contact with siVEGFR3 cells was significantly higher ($96.9 \pm 6.9\%$, $n = 5$) than expected, if the proliferating cells were randomly distributed (Fig. 5f). Proliferation of siCTRL cells was not affected when co-cultured without a direct contact with siVEGFR3 cells seeded on a Transwell culture insert (Fig. 5g). These results demonstrate that VEGFR3-deficient LECs induce proliferation of neighboring WT LECs in a cell-contact-dependent manner.

**Low DLL4 in VEGFR3$^-$ LECs promotes proliferation in neighbors.** DLL4-Notch signaling is an important cell-contact-mediated regulator of sprouting (lymph)angiogenesis[4]. In the blood vasculature, DLL4 is induced by VEGF and VEGF-C in

ECs at the tips of vessel sprouts[7,28,29]. DLL4 activates Notch in neighboring cells, which in most EC types and vascular beds has been shown to result in cell cycle arrest[11,30,31,32]. Conversely, inhibition of DLL4 or NOTCH1 leads to excessive blood and lymphatic vessel sprouting and EC proliferation[28,29]. Interestingly, in the Drosophila tracheal system, which develops via mechanisms analogous to the vertebrate vascular system, Notch signaling additionally specifies cells at the tips of the developing tracheal branches by repressing 'fusion cells' (cells mediating branch fusion) and promoting 'terminal cells' (cells extending terminal branches) fate[33–35,36]. We therefore hypothesized that Notch signaling contributes to both excessive vascular anastomosis and proliferation of VEGFR3$^+$ LECs that are in contact with VEGFR3$^-$ LECs. In agreement with this hypothesis, quantitative real-time PCR (qRT-PCR) analysis of human siRNA-treated LECs revealed significant downregulation of *DLL4* in *VEGFR3*-deficient cells (Fig. 6a). The levels of *NOTCH1* were modestly lower, and the *NOTCH1* target *HEY1* was strongly reduced, reflecting decreased Notch activation due to low DLL4 expression in these cells (Fig. 6a). Analysis of mouse LECs isolated from the *Vegfr3$^{flox/flox}$;R26-mTmG;Prox1-CreER$^{T2}$* ears confirmed *Dll4* downregulation but unchanged *Notch1* expression in the *Vegfr3* deficient in comparison to Cre$^-$ control LECs (Fig. 6b). Whole-mount immunofluorescence further confirmed downregulation of DLL4 protein specifically in GFP$^+$ (i.e., VEGFR3$^-$) LECs in comparison to non-targeted GFP$^-$ (i.e., VEGFR3$^+$) LECs in the mutant ears (Fig. 6c). Further analysis of cells in mosaically targeted vasculature in vivo showed that Notch signaling, as assessed by *Hey1* expression, was downregulated in both non-targeted VEGFR3$^+$ (GFP$^-$Tomato$^+$) LECs and targeted VEGFR3$^-$ (GFP$^+$Tomato$^-$) LECs from the *Vegfr3$^{flox/flox}$; R26-mTmG;Prox1-CreER$^{T2}$* mutant ears compared to LECs from Cre$^-$ control mice (Fig. 6d). In addition, *Efnb2*, a previously demonstrated target of DLL4-Notch1 signaling in the lymphatic vasculature[20], was strongly downregulated in the non-targeted LECs (Fig. 6e). Higher expression of the proliferation marker *Ccnb1* confirmed increased proliferation of the non-targeted LECs (Fig. 6f).

To further investigate if LEC proliferation is affected by decreased DLL4-Notch signaling, we cultured human LECs under confluent growth-inhibiting conditions in the presence of low (25 ng ml$^{-1}$) VEGF-C stimulation. Wild-type LECs showed a low proliferation rate, which was further reduced upon VEGFR3 silencing (Fig. 7a, b). Inhibition of Notch signaling by DAPT treatment increased proliferation of wild type but not VEGFR3$^-$ LECs (Fig. 7a, b). Similarly, LECs treated with *NOTCH1* siRNA (siNOTCH1) (Fig. 7c) showed increased proliferation in comparison to siCTRL cells (Fig. 7d). Although Notch inhibition promoted proliferation in LECs cultured in complete serum containing medium in the absence of VEGF-C, the effect was more pronounced when VEGF-C was present (Fig. 7d), suggesting the importance of VEGF-C-VEGFR3 signaling in this response. To investigate if mosaic loss of Notch activation is sufficient to phenocopy the effect of mosaic loss of VEGFR3 on the proliferation of the neighboring cells, we performed co-culture experiments using LECs treated with siCTRL or siRNA against *DLL4* (siDLL4) (Fig. 7e). As expected, siDLL4 LECs showed increased proliferation in comparison to siCTRL cells when cultured alone (Fig. 7f). In addition, increasing the proportion of siDLL4 cells in the co-cultures increased the proliferation of siCTRL LECs (Fig. 7f).

Together, the above results demonstrate that VEGFR3 mediated regulation of DLL4 controls Notch induced cell cycle arrest in the LECs in a non-cell-autonomous manner. LECs with low Notch respond by proliferation but only in the presence of VEGFR3 (Fig. 7g).

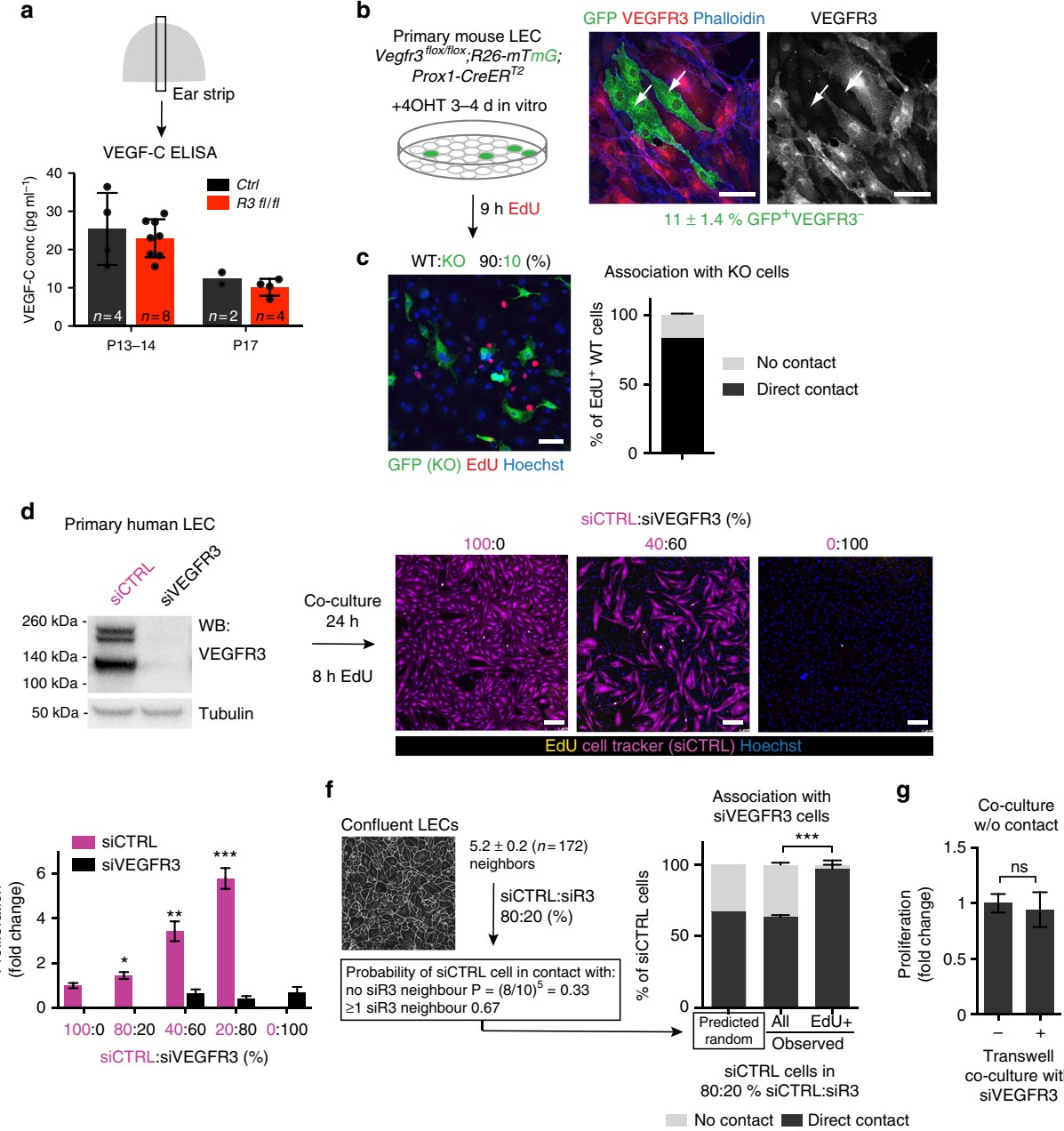

**Fig. 5** *Vegfr3*-deleted LECs induce proliferation of neighboring VEGFR3+ cells in a cell-contact-dependent manner. **a** VEGF-C concentration measured by ELISA in the ear skin at indicated stages. Bars represent mean (*n* = 2–8 mice, as indicated) ± s.d. **b** Experimental set up for co-cultures of WT and VEGFR3⁻ (KO) primary LECs from *Vegfr3^flox/flox^;R26-mTmG;Prox1-CreER^T2^* mice (left) and immunofluorescence showing VEGFR3 depletion in Cre⁻ targeted (GFP⁺) LECs (right, arrows). **c** Left: proliferating WT cells were frequently associated with KO cells. Right: quantification of the proportion of EdU⁺ WT cells that were in direct contact with KO cells (mean (*n* = 6 biological replicates) ± s.e.m). **d** Assessment of cell proliferation in co-cultures of primary human LECs treated with control (siCTRL) or VEGFR3 (siVEGFR3) siRNA. Cell tracker-labeled siCTRL cells (purple) were mixed with unlabeled siVEGFR3 cells in 40:60%, or they were cultured alone. **e** Quantification of proliferating EdU⁺ cells in co-cultures of siCTRL and siVEGFR3 LECs mixed in different ratios. Data are normalized to the proliferation rate (0.8 ± 0.2%) in siCTRL alone and represent mean (*n* = 5 biological replicates) ± s.e.m. **f** Predicted and observed proportions of siCTRL LECs (all vs. EdU⁺) that are in direct contact with siVEGFR3 LECs in 80:20% siCTRL:siVEGFR3 co-cultures (mean (*n* = 5 biological replicates) ± s.e.m). Predicted proportion was calculated based on random distribution and *n* = 5 (5.2 ± 0.2, quantified from *n* = 172 LECs stained for VE-cadherin) neighbors. **g** Quantification of proliferating EdU⁺ cells in siCTRL LECs cultured alone (−), or co-cultured without direct cell–cell contacts with siVEGFR3 cells seeded on Transwell inserts (+). Data are normalized to the proliferation rate in siCTRL alone and represent mean (*n* = 6 biological replicates) ± s.e.m. *P < 0.05, **P < 0.01, ***P < 0.001. Two-tailed unpaired Student's *t* test (**a, e, g**) and Fisher's exact test (**f**). Scale bars: 50 μm (**b, c**), 200 μm (**d**). ns: not significant

## Discussion

VEGF-C signaling through its tyrosine kinase receptors VEGFR2 and VEGFR3 is the key regulator of embryonic and post-natal lymphatic development[14,17,22]. Here we show, using conditional

Cre/loxP mediated gene deletion, that VEGFR3 is the major VEGF-C receptor during embryonic lymphangiogenesis. LEC-specific deletion of *Vegfr3* in embryos led to dermal lymphatic vessel hypoplasia and revealed an indispensable function of

VEGFR3 for lymphatic endothelial tip cell function and vessel sprouting. Our data further demonstrate that the embryonic requirement of VEGFR3 for LEC survival and proliferation is conserved in neonatal ear vasculature. Yet LEC-specific deletion of VEGFR3 within the critical post-natal period of 2 weeks led to dermal lymphatic vessel hyperplasia. Investigation of the

underlying cause of this unexpected phenotype uncovered a non-cell-autonomous mechanism by which VEGFR3-deleted cells induce excessive proliferation of residual VEGFR3$^+$ LECs that escaped Cre recombination.

In blood vascular ECs, loss of VEGFR3 results in increased VEGFR2 levels and signaling[7,23], which was shown to drive

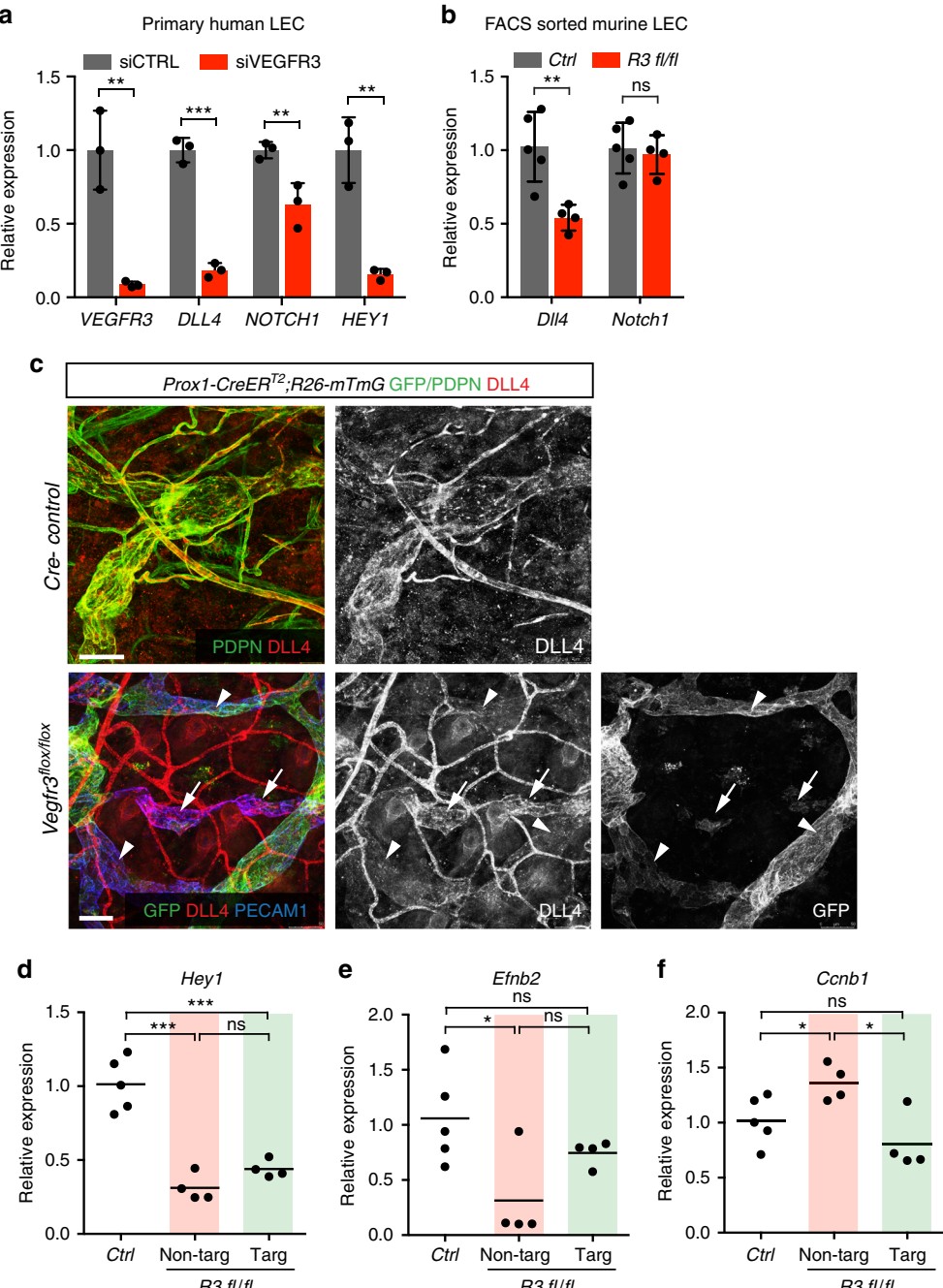

**Fig. 6** DLL4 downregulation and decrease in Notch signaling upon loss of VEGFR3. **a** qRT-PCR analysis of Notch pathway genes in control (siCTRL) and VEGFR3 siRNA (siVEGFR3)-treated human LECs. Bars represent mean relative expression ($n = 3$ biological replicates with $n = 2$ technical replicates each) ± s.d. **b** qRT-PCR analysis of *Dll4* and *Notch1* expression in FACS-sorted LECs from P13-P14 Cre$^-$ (*Ctrl*) and *Vegfr3$^{flox/flox}$;R26-mTmG;Prox1-CreER$^{T2}$* (*R3 fl/fl*) mice. Tamoxifen (150 μg) was administered at P2, P4, and P6. Bars represent mean relative expression ($n = 5$ *Ctrl* and $n = 4$ *R3 fl/fl* mice) ± s.d. **c** Whole-mount immunofluorescence of ear skin of 3 weeks old *Vegfr3$^{flox/flox}$;R26-mTmG;Prox1-CreER$^{T2}$* and Cre-negative littermate mice treated with Tamoxifen at P2, P4, and P6. Note downregulation of DLL4 protein in targeted GFP$^+$ LECs (arrows) compared to non-targeted GFP$^-$ LECs (arrowhead) in the mutant (*flox/flox*) ear. **d** *Hey1*, (**e**) *Efnb2* and (**f**) *Ccnb1* expression analyzed by qRT-PCR in FACS-sorted LECs from P13-P14 Cre$^-$ (*Ctrl*) and *Vegfr3$^{flox/flox}$;R26-mTmG; Prox1-CreER$^{T2}$* (*R3 fl/fl*) mice. Results from non-targeted (Tomato$^+$GFP$^-$; red), and targeted (Tomato$^-$GFP$^+$; green) LECs are displayed separately. Horizontal lines represent mean relative expression ($n = 5$ *Ctrl* and $n = 4$ *R3 fl/fl* mice with $n = 2$ technical replicates for each). *$P < 0.05$, **$P < 0.01$, ***$P < 0.001$. Two-tailed unpaired Student's *t* test (**a**, **b**, **d**–**f**). Scale bars: 50 μm (**c**). ns: not significant

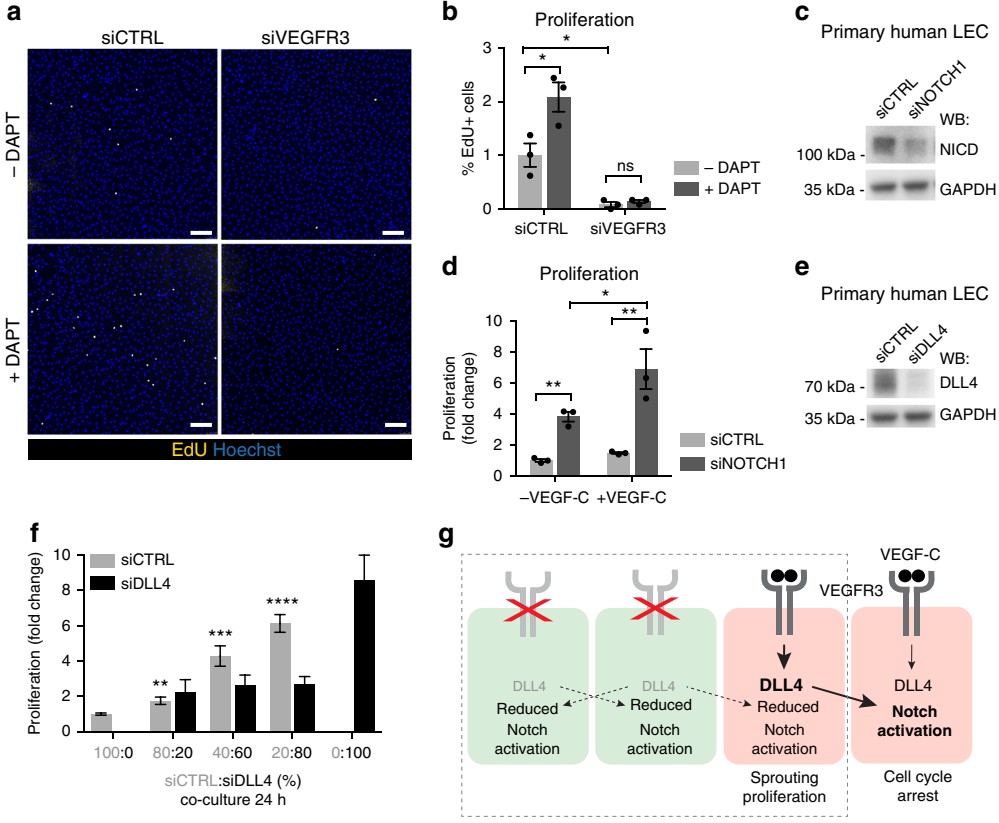

**Fig. 7** Global and mosaic loss of DLL4 promote LEC proliferation through inhibition of Notch signaling in neighboring cells. **a**, **b** DAPT induced effect on the proliferation of control (siCTRL) or VEGFR3 (siVEGFR3) siRNA-treated LECs. **a** Tile scan immunofluorescence images and (**b**) quantification of proliferating EdU$^+$ LECs after 3 h of incorporation (mean ($n = 3$ biological replicates) ± s.e.m). **c** Western blot analysis of cell lysates from LECs treated with siCTRL or NOTCH1 siRNA (siNOTCH1) and (**d**) quantification of proliferating EdU$^+$ LECs cultured with or without VEGF-C (data are normalized to the proliferation rate in siCTRL alone and represent mean ($n = 3$ biological replicates) ± s.e.m). **e** Western blot analysis of cell lysates from LECs treated with siCTRL or DLL4 siRNA (siDLL4) and (**f**) quantification of proliferating EdU$^+$ cells in co-cultures of siCTRL and siDLL4 LECs mixed in different ratios (data are normalized to the proliferation rate in siCTRL alone and represent mean ($n = 6$ biological replicates) ± s.e.m). **g** Model of cell-autonomous and non-cell-autonomous effects of VEGFR3 expression defining LEC responses during sprouting lymphangiogenesis. VEGF-C-VEGFR3 signaling positively regulates DLL4 expression leading to activation of Notch and cell cycle arrest in neighboring LECs. VEGFR3 deficiency (green cell) consequently leads to downregulation of DLL4 and inhibition of Notch in neighboring LECs, but only VEGFR3 expressing (red) cells respond by proliferation and sprouting. When VEGFR3$^-$ cells are in excess, VEGFR3$^+$ cells interact predominantly with LECs with low DLL4 and respond by proliferation (dashed line) *$P < 0.05$, **$P < 0.01$, ***$P < 0.001$, ****$P < 0.0001$. Two-tailed unpaired Student's $t$ test (**b**, **d**, **f**) or one-way ANOVA with Tukey's post hoc test (**d**; comparison between VEGF-C untreated and treated groups). Scale bars: 200 μm (**a**). ns: not significant

hyperplasia of retinal vasculature in vivo upon genetic deletion of *Vegfr3*[7]. We found that the mechanism underlying lymphatic vessel hyperplasia upon LEC-specific *Vegfr3* deletion is different. Namely, VEGFR2 levels were not altered in the *Vegfr3*-deleted LECs, and LEC-specific genetic deletion of *Vegfr2* did not rescue lymphatic vessel hyperplasia induced by *Vegfr3* loss. In contrast, dermal lymphatic vessel hyperplasia was prevented by global inhibition of VEGF-C signaling using the VEGF-C-trap, indicating the dependence of the phenotype on VEGF-C signaling in the non-targeted VEGFR3$^+$ LECs. Global inhibition of VEGFR3 activity using the MAZ51 tyrosine kinase inhibitor in the same setting inhibited abnormal lymphatic vessel anastomosis, while inhibition of VEGFR2 function using the blocking antibody DC101 reduced vessel hyperbranching. These data suggest specific and synergistic functions of VEGFR2 and VEGFR3 in driving the abnormal VEGF-C-driven behavior of the non-targeted VEGFR3$^+$ LECs that are in contact with VEGFR3$^-$ LECs.

Our results should, however, be interpreted with some caution since it is not possible to ensure complete or equal inhibition of the two receptors with the inhibitors used. That MAZ51 did not have an apparent effect on wild-type vasculature at P21 indeed

suggests that the inhibitor does not fully inhibit VEGFR3 activity. Another consideration is that while MAZ51 targets only the non-recombined VEGFR3$^+$ cells in the mutant skin, VEGF-C-trap and DC101 target both the recombined VEGFR3$^-$ and non-recombined VEGFR3$^+$ LECs by inhibiting VEGFR2 signaling. Although LEC-specific deletion of *Vegfr2* alone or in combination with *Vegfr3* did not indicate a function for VEGFR2 in the LECs, DC101 slightly reduced lymphatic growth in wild-type mice. It is thus not possible to exclude a minor role for VEGFR2 in lymphatic development or in the VEGFR3-deleted LECs, because of, e.g., slow kinetics of VEGFR2 depletion in the genetic mutants.

Post-natal LEC-specific genetic deletion of *Vegfr3* led to hyperplasia of lymphatic vasculature both in the skin and in the intestinal wall. The process by which VEGFR3$^-$ LECs non-cell-autonomously regulate lymphatic vessel growth by inducing proliferation of non-targeted VEGFR3$^+$ LECs through cell-contact-dependent reduction in Notch signaling is however unlikely to occur in every tissue with active VEGF-C-dependent lymphangiogenesis. It is possible that lymphatic vascular beds that form through a plexus intermediate via repeated cycles of vessel sprouting and anastomosis are particularly sensitive to aberrations in Notch signaling, based on the established function

of this pathway in controlling similar processes in the blood vasculature[4,37]. Intriguingly, previous studies have revealed context-dependent roles of Notch signaling in the lymphatic vasculature, whereby inhibition of Notch signaling can have opposite effects depending on the tissue and developmental stage. For example, inhibition of DLL4-Notch1 signaling inhibits dermal lymphatic vessel growth and sprouting in neonatal mice[20], but promotes vessel hyperplasia in adult mouse skin[29]. Interestingly, inhibition of VEGF-C signaling blocks lymphatic vessel growth and induces regression during embryonic and ear post-natal development, but not after 2 weeks of age[16,22]. Our data showing that VEGFR3 is an upstream regulator of DLL4 is thus consistent with the early dependence of lymphatic growth on DLL4-Notch1 signaling. The later switch of LECs to become independent of VEGF-C for their survival, with the exception of meningeal lymphatic vessels[38], may be due to changes in matrix-dependent signals or processes involved in vascular plexus formation (initial plexus formation vs. remodeling). It should be noted that different approaches of inhibiting pathway activity (genetic deletion vs. soluble inhibitor; mosaic vs. global) may result in different outcomes, which makes comparison of phenotypes difficult.

Our results do not fully explain why Vegfr3 deletion induces lymphatic vessel hyperplasia and hyperproliferation of non-targeted LECs in early post-natal skin but not in embryonic skin. Different kinetics of vascular plexus formation, but also the kinetics of gene deletion[18,21] in the two situations (4-OHT—fast kinetics in embryos vs. Tamoxifen—slower kinetics in neonatal mice) may affect the outcome that is determined by a balance between cell-autonomous and non-cell-autonomous effects of Vegfr3 deletion. In the embryo the primitive dermal vascular plexus forms within 2–3 days, but in the post-natal ear, this process takes 10–12 days. It is possible that during the rapid expansion of embryonic vasculature the cell-autonomous effect of Vegfr3 deletion on LEC proliferation and survival dominates. Another possibility is that a specific subset of LECs in the early post-natal vasculature are not efficiently targeted by the Prox1-CreER[T2], allowing for a sufficient number of LECs to escape recombination and respond to VEGFR3[−] neighbors by proliferation. The observed efficient Prox1-CreER[T2] driven recombination in wild-type vasculature does not suggest that this is the case, however, such a scenario may become evident only when non-targeted cells show a selective growth advantage.

The established view on sprouting angiogenesis places tip cells exhibiting high VEGFR2 activation and DLL4 expression as the initiators of the molecular and cellular processes of normal and pathological vessel sprouting. We show that LECs with low VEGFR3 are capable of promoting the formation of new vessel sprouts and severe lymphatic vessel hyperplasia by stimulating proliferation and vascular anastomosis in neighboring LECs in a cell-contact-dependent manner. In addition to providing inductive signals for vascular morphogenesis, such an unanticipated non-cell-autonomous mechanism of driving tissue growth may contribute to hyperplastic pathologies in the vasculature and in other tissues. For example, it is not known if ECs carrying somatic mutations underlying vascular malformations can engage normal ECs for growth during lesion formation through non-cell-autonomous mechanisms, but the reported low frequency of mutant cells present in the affected tissue suggests this possibility[39].

The phenomenon we describe here by which a mosaic loss/inhibition of a growth promoting pathway can unexpectedly promote tissue growth through non-cell-autonomous effects can be envisaged to apply to any tissue and growth promoting signaling pathway. Recently, such a concept has emerged in cancer, where interactions between different subclones of tumor cells

promote tumor progression[40,41]. It is also recognized that this clonal cooperation can have a profound effect on therapeutic outcomes, thus the identification of non-cell-autonomous driver subclones that promote tumor growth by influencing nearby populations is critical. Since a major problem of therapeutic approaches such as siRNA nanovectors[42,43] is an incomplete delivery to the target cell population, leading to generation of subclones with heterogeneous expression of the target gene, understanding of non-cell-autonomous effects will be an important consideration for future studies.

Our data highlight challenges with interpreting phenotypes from conditional Cre/loxP mouse models. When the studied gene is critically required for a particular cellular process, cells that escape Cre recombination obtain a selective advantage. The observation that gene-deleted cells are not competent to contribute to a particular process or tissue structure may reveal indispensable functions for proteins. This is demonstrated by the failure of ECs lacking NRP1[44] or VEGFR3 (this study) to lead the sprouts of blood or lymphatic vessels, respectively. However, if the contribution of non-recombined cells selectively in the mutant tissues is not recognized, gene functions may be overlooked. Our study also demonstrates a new concept that Cre-targeted cells can drive abnormal processes through non-cell-autonomous effects. Repeated induction of Vegfr3 deletion in a new pool of non-targeted cells was critical for the development of lymphatic hyperplasia in the present study. This can be explained by the need for direct cell–cell interactions between targeted and non-targeted cells, which after a pulsed Cre induction are not expected to be maintained during the dynamic sprouting process due to the selective survival, expansion, and migratory advantage of the non-targeted cells. It is important to note that even in the absence of repeated administrations of Tamoxifen, high serum levels of the active metabolite 4-OHT are maintained for up to 72 h, compared to a faster 24 h clearance of administered 4-OHT[18,21]. Depending on the dose, recombination has been shown to occur weeks after the last Tamoxifen administration[45]. Therefore, characterization of Cre-dependent phenotypes should ideally take into account the kinetics of gene deletion (and protein depletion), genotype-specific cell selection, and non-cell-autonomous effects that may be at play.

In conclusion, we show that VEGFR3 regulates sprouting lymphangiogenesis through cell-autonomous and non-cell-autonomous mechanisms. Unexpectedly, VEGFR3[−] LECs non-cell-autonomously drive abnormal lymphatic vessel anastomosis and hyperplasia in LEC-specific Vegfr3 knock-out mice by lateral induction of cell proliferation and vessel anastomosis in the neighboring VEGFR3[+] LECs that escaped Cre recombination. The finding that heterogeneous inhibition of a growth promoting signaling pathway promotes vascular hyperplasia has important implications for the understanding of mechanisms of developmental and pathological tissue growth.

## Methods

**Mice**. R26-mTmG mice[46] were obtained from the Jackson Laboratory. Prox1-CreER[T2][47], Vegfr3[flox][48], and Vegfr2[flox][49] lines were described previously. Vegfr2[+/−] mice were generated by crossing Vegfr2[flox] with the PGK-Cre mice, followed by removal of the Cre transgene by crossing with C57BL/6J mice. All mice were maintained on a C57BL/6J genetic background. For staging of embryos, the morning of vaginal plug detection was considered E0. For induction of Cre recombination, 4-hydroxytamoxifen (4-OHT), dissolved in peanut oil (10 mg ml[−1]), was administered to pregnant females by intraperitoneal injections as indicated in figures and/or legends. For neonatal deletion, Tamoxifen or 4-OHT, dissolved in acetone (10 mg ml[−1]), was applied topically to abdominal skin using the indicated doses and regimes. Alternatively, 4-OHT dissolved in ethanol (25 mg ml[−1]) was given intraperitoneally (2 μl). For AAV transduction, pups were injected intraperitoneally at P10 with $5 \times 10^{10}$ viral particles in 8.4 μl of a recombinant AAV9 encoding the ligand binding domains 1–4 of VEGFR3, fused to the IgG Fc domain (AAV9-mVEGFR3$_{1-4}$-Ig;[24]). For treatment with blocking antibody or

inhibitor, pups were injected intraperitoneally every second day from P7 to P17 with 40 mg kg$^{-1}$ VEGFR2 blocking antibody DC101 (BE0060, BioXcell) or 10 mg kg$^{-1}$ VEGFR3 inhibitor MAZ51 (676492, Calbiochem). To assess cell proliferation in vivo, pups were administered with 100 μl of EdU (5 mg ml$^{-1}$) by subcutaneous injection using the indicated regimes. Crosses were performed until a minimum of three control or mutant animals from a minimum of two litters were obtained. Experimental procedures were approved by the Uppsala Laboratory Animal Ethical Committee.

**Primary LEC culture.** Human primary dermal lymphatic ECs (HDLEC from juvenile foreskin, cat. no. C12216) were obtained from PromoCell. Cells were seeded on bovine Fibronectin (F1141, Sigma) coated dishes in complete ECGMV2 medium (PromoCell) supplemented with 25 ng ml$^{-1}$ of VEGF-C (2179-VC, R&D Systems) and used after three to four passages. For assessing proliferation, cells were cultured under confluent contact-inhibited conditions ($6.3 \times 10^4$ cells per cm$^2$). Murine primary dermal lymphatic ECs were isolated from the tail skin of 6 weeks old $Vegfr3^{flox/flox};R26\text{-}mTmG;Prox1\text{-}CreER^{T2}$ female by sequential selection with PECAM1 (Mec13.3, Pharmingen) and LYVE1 (Aly7, Abnova) antibodies bound to Dynabeads, as described previously[50]. Mouse LECs were cultured on 0.5% gelatin-coated dishes and passaged 1:3 after 4 days. In vitro gene deletion was induced by adding 1 μM 4-OHT (H7904, Sigma) to the culture medium. Notch signaling was inhibited by 24 h treatment with 10 μM DAPT (CAS 208255-80-5, Calbiochem). All cells were cultured at 37 °C in a humidified atmosphere with 5% CO$_2$.

**RNA interference.** Human dermal LECs were transfected with siRNA against VEGFR3 (SI02225454, Qiagen), NOTCH1 (SI00119028, Qiagen), DLL4 (L-010490, Dharmacon), or control siRNA (1027281, Qiagen) using Lipofectamine 2000 (Invitrogen) for 12 h before replacing the transfection medium with culture medium. For VEGFR3 siRNA, cells were re-transfected after 12 h followed by re-seeding or collecting 12 h later.

**Immunofluorescence.** Whole-mount immunofluorescence was done as previously described[18,19] with minor modifications. Two dermal sides of ears were separated by forceps and the dorsal side was processed for staining. A small piece of intestine (jejunum) was carefully collected and pinned down with outer side upward. First, tissue was fixed in 4% paraformaldehyde (PFA), permeabilized in 0.3% Triton X-100 in PBS (PBST), and blocked in PBST plus 3% milk. Primary antibodies were incubated at 4 °C overnight in blocking buffer. After washing in PBST, the samples were incubated with fluorescence-conjugated secondary antibodies in blocking buffer before further washing and mounting in Mowiol. Cells grown on 48-well plate were fixed with 4% PFA at room temperature (RT) for 20 min, washed twice with PBS and permeabilized in 0.5% Triton X-100 in PBS for 5 min, and blocked in PBST plus 2% BSA for 1 h. Cells were incubated with primary antibodies diluted in the blocking buffer at RT for 1 h. After washing three times with PBS + 0.2% BSA, the cells were incubated with fluorescence-conjugated secondary antibodies at RT for 45 min, followed by staining with DAPI (D9542, Sigma, 1:1000) for 5 min, and fixing with 4% PFA for 4 min before further washing and mounting in Mowiol. The following antibodies were used for immunofluorescence: chicken anti-GFP (ab13970, Abcam, 1:500), hamster anti-mouse Podoplanin (clone 8.1.1, Developmental Studies Hybridoma Bank, 1:200), rat anti-mouse PECAM1 (553370, BD Biosciences, 1:1000), rabbit anti-mouse LYVE1 (103-PA50AG, Reliatech, 1:200), α-smooth muscle actin (SMA)-Cy3 (C6198, Sigma, 1:300), rat anti-mouse Endomucin (sc-65495, Santa Cruz, 1:200), rat anti-mouse F4/80 (MCA497GA, AbD Serotec, 1:200), rat anti-mouse LYVE1 (MAB2125), goat anti-mouse Nrp2 (AF567), goat anti-mouse DLL4 (AF1389), goat anti-mouse VEGFR2 (AF644), goat anti-mouse VEGFR3 (AF743; all from R&D Systems and 1:200). Secondary antibodies conjugated to Cy3, AlexaFluor 488 or 647 were obtained from Jackson ImmunoResearch (1:300). To assess cell proliferation in vitro, Click-iT EdU Alexa Fluor 647 Imaging kit (C10340, Thermo Fisher Scientific) was used. Primary mouse or human LECs were incubated with a final concentration of 10 μM EdU at 37 °C for indicated time periods, followed by washing and EdU detection according to manufacturer's instructions.

**Flow cytometry.** Ear skin was digested in 5–10 mg ml$^{-1}$ Collagenase IV (Life Technologies) and 0.2 mg ml$^{-1}$ DNase I (Roche) in PBS plus 0.5% fetal calf serum (FCS) (Gibco) at 37 °C for 20–30 min. Digested samples were quenched by adding 2 mM EDTA and filtered through a 70 μm nylon filter (BD Biosciences). Cells were washed with FACS buffer (PBS, 0.5% FCS, 2 mM EDTA) and immediately processed for staining in 96-well plates. Fc receptor binding was blocked by rat anti-mouse CD16/CD32 (93) (eBioscience, 1:100). For RNA cell sorting, ECs were enriched using CD31/PECAM1 magnetic microbeads (Miltenyi Biotech) on a MS column according to the manufacturer's instructions. Cells were stained using a combination of anti-CD31/PECAM-1 (390) PE-Cy7 (1:300), anti-podoplanin (PDPN) (eBio8.1.1) eF660 (1:100), and biotinylated anti-VEGFR2 (Avas12a1) (1:50) (all from eBioscience). VEGFR2 was detected by adding SA-PERCP-Cy5.5 (eBioscience, 1:300) in a second staining step. Dump channel included markers to exclude immune cells using anti-CD45 (30-F11); myeloid cells using anti-CD11b (M1/70) and red blood cells using anti-TER-119 (TER-119) antibodies; all diluted

1:50 and conjugated to eF450 (eBioscience); together with Sytox blue (Life Technologies) for dead cell exclusion. LECs were gated in three steps; (1) PECAM1$^{high}$, dump channel$^-$ cells, (2) PECAM1$^+$, VEGFR2$^+$ (ECs), (3) PDPN$^+$ (LECs). LECs were subsequently sorted based on GFP and Tomato expression. Sorting was performed on a FACS Aria III with the FACSDiva software (BD Biosciences) using a 100 μm nozzle, 20–25 pounds per square inch (psi), with a highly pure sorting modality (four-way purity sorting), and an acquisition rate of 500–1000 events per s. A total of 500–1000 cells were sorted directly into RLT buffer with beta-mercaptoethanol and immediately processed for RNA extraction using the Qiagen micro kit plus. To assess proliferation, cells were stained with fluorophores compatible with the Click-it EdU AF647 flow cytometry kit (Thermo Fisher Scientific) in five steps: (1) anti-CD45 (30-F11) (1:50) and anti-CD11b (M1/70) (1:50) both conjugated with Percp-Cy5.5 and anti-CD31/PECAM-1 (390) BV510 (1:50); (2) LIVE/DEAD fixable near-infrared dead cell stain kit (Invitrogen); (3) fixation and Click-it reaction according to the manufacturer's instructions; (4) chicken anti-GFP (13970, Abcam, 1:200); (5) donkey anti-chicken AF488 (Jackson ImmunoResearch, 1:300) and rat anti-mouse PDPN (eBio8.1.1) PE-Cy7 (1:300). LECs were gated in three steps: (1) PECAM1$^+$, cell death$^-$ (live ECs); (2) PECAM1$^{high}$, CD45/CD11b$^-$ (EC pure); (3) PDPN$^+$ (LEC). LECs were further analyzed for GFP and Tomato expression and EdU labeling. In some experiments, an additional staining step for VEGFR2 was included as described for the sorting of cells before fixation and Click-it reaction. In this case, fluorophores for detection of CD11b and CD45 were exchanged to AF450. Data were processed using FlowJo software (TreeStar). Single cells were gated using FSC-A/SSC-A followed by FSC-H/FSC-W and SSC-H/SSC-W in all experiments. The anti-rat/hamster compensation bead kit (Thermo Fisher Scientific) was used for compensation controls with addition of GFP$^+$, Tomato$^+$, and LIVE/DEAD fixable near-IR dead cell stained cells.

**Image acquisition and quantification.** Images were acquired using Zeiss LSM 780 confocal microscope and Zen software (Fig. 1b), or Leica SP8 confocal microscope with HC PL APO CS 10×/0.4 DRY or HL PL APO 40×/1.1 W motCORR CS2 objective and LAS X software. All confocal images represent maximum intensity projections of Z-stacks of single tile or multiple tile scan images.

Quantification of lymphatic vessel parameters (branching, diameter, tip cell occupancy, number of cells and blunt-ended vessels, avascular midline area, vessel area) was done using maximum intensity projection images of tile scanned E17.5 skins ($xy = 3400 \times 1700$ μm, upper thoracic region); P14, P21 or 5-week ear skin ($xy = 3260 \times 2200$ μm at the tip of the ear) or P21 intestinal wall ($xy = 2050 \times 2050$ μm); $n = 3$–10 mice as indicated in the figures and/or figure legends. All branch points in each image were marked using Photoshop CS6 software and counted manually. Vessel diameter was determined by measuring the thickest part of each vessel segment in between branch points ($n = 30$ measurements per skin) and mean value was plotted for each embryo. For measurement of the distance of lymphatic sprouts from the midline (i.e., avascular midline), dorsal midline was marked with a line and six to eight measurements, three to four on each side of the midline, were taken from each image using the LSM image browser (Zeiss) and mean value was plotted for each embryo. Lymphatic vessels closest to the midline were included. GFP$^+$ cells in the midline (area depicted in Fig. 1i) were counted manually. The distinction between individual GFP$^+$ cells was made based on staining with nuclear marker PROX1 or membrane-bound GFP signal marking the borders between adjacent GFP$^+$ cells. Blunt-ended vessels were counted manually in the tip (560 μm from the edge of the ear) and central regions of the ear. For quantification of tip cell occupancy by Cre-targeted cells, presence of GFP$^+$ cells were counted manually in vessel tips visualized by PDPN (ear) and Nrp2 or VEGFR3 (embryonic skin) staining using Image J or Adobe Photoshop software. In the ear, lymphatic vessels closest to the edge of the ear tip were included. Vessel area was determined as % of LYVE1$^+$ or VEGFR3$^+$ area of the total area using Image J software. All quantifications were done with littermate controls, which were either heterozygous $Vegfr3^{flox/+};Prox1\text{-}CreER^{T2}$ mice or Cre-negative $Vegfr3^{flox/flox}$ or $Vegfr3^{flox/+}$ mice as indicated in the figures and/or figure legends. In some experiments, wild-type controls were additionally generated as separate litters of $R26\text{-}mTmG;Prox1\text{-}CreER^{T2}$ (Fig. 1c–g) or C57BL/6J (Fig. 2c, e, f, Fig. 3a, and Supplementary Fig. 4c) mice.

Quantification of proliferation and association of WT LECs with $Vegfr3$ KO LECs (mouse), and siCTRL with siVEGFR3/siDLL4/siNOTCH1 LECs (human) in vitro was done using maximum intensity projection images of tile scans ($xy = 1720 \times 1720$ μm). Cell proliferation was determined as % EdU$^+$ cells of all Hoechst$^+$ nuclei using Image J software. To assess association of proliferating WT/siCTRL LECs with $Vegfr3$ KO/ siVEGFR3/ siDLL4 LECs, the former were labeled with cell tracer Green CMFDA dye according to the manufacturer's instructions (C2925, Thermo Fisher Scientific) 24 h before co-culture. All EdU$^+$ and Hoechst$^+$ WT cells that had a direct contact with $Vegfr3$ KO/siVEGFR3/siDLL4 cells were marked using Photoshop CS6 software and counted manually.

**Western blot.** LECs were washed with PBS and scraped from the dish with heated lysis buffer (70 mM Tris-HCl, pH 6.8, 4.3% SDS, 14.5% glycerol), and homogenized with a syringe (1 ml with a 27 G needle) for eight times. After centrifugation, supernatants were collected followed by addition of bromophenol blue and 5% β-mercaptoethanol and boiling at 95 °C for 5 min. Equal amounts of total cell lysates

were separated in 4-20% gradient polyacrylamide gels (Invitrogen), transferred to polyvinylidene difluoride membrane (PVDF; Thermo Fisher Scientific) and probed with antibodies as indicated. Proteins were detected using mouse anti-human VEGFR3 (clone 9D9, MAB3757, Millipore, 1:50), rabbit anti-human Cleaved NOTCH1 (Val1744) (#4147, CST, 1:1000), rabbit anti-human DLL4 (#2589, CST, 1:1000), rabbit anti-human GAPDH (#2118, CST, 1:3000) and mouse anti-human α-tubulin (T5168, Sigma, 1:1000) and visualized by using an ECL chemiluminescent substrate reagent (Invitrogen). Full blots are shown in Supplementary Fig. 8.

**qRT-PCR analysis**. Total RNA was isolated from human dermal LECs using RNeasy Mini kit (Qiagen) and from sorted LECs using RNeasy Micro plus kit (Qiagen). Complementary DNA (cDNA) was synthesized using Superscript VILO Master Mix (Invitrogen). cDNA was pre-amplified using TaqMan PreAmp Master Mix (Applied Biosystem), and analyzed by real-time quantitative PCR (qRT-PCR) (StepOne Plus system, Applied Biosystems) using TaqMan gene expression assays and TaqMan gene expression Master Mix (Applied Biosystems). Relative quantifications of gene expression were performed using the comparative cycle threshold method (ΔCT) with *GAPDH* as the reference gene. TaqMan Assays used for human LECs were as followed: *VEGFR3* (Hs00176607_m1), *DLL4* (Hs00184092_m1), *NOTCH1* (Hs01062014_m1), *HEY1* (Hs01114113_m1), *GAPDH* (Hs02786624_g1). TaqMan assays used for mouse LECs were: *Vegfr2* (Mm01222421_m1), *Vegfr3* (Mm01292604_m1), *Dll4* (Mm00444619_m1), *Notch1* (Mm00627185_m1), *Efnb2* (Mm00438670_m1), *Hey1* (Mm00468865_m1), *Ccnb1* (Mm03053893_gH), *Vegfc* (Mm00437310_m1), *Vegfd* (*Figf*) (Mm01131929_m1), *Gapdh* (Mm99999915_g1). The values represent average relative gene expression. The average ΔCT value for control LECs was used for normalization unless specified differently.

**Genotyping of sorted LECs**. Primers and probe covering exon 1 of *Vegfr3* were designed: exon1F1: GCTGAACCTGCGCCTGTG; exon1R1: TTAAGCGG-GAGCGGAGGTTG and probe exon1P1: TGAGCGCCCGGGCCACAGCC, giving an amplicon size of 82 bp. The probe was labeled with 5′Fam–3′Tamra. A total of 100–300 LECs were sorted by flow cytometry (as described above) into lysis buffer containing 25 mM NaOH and 0.2 mM EDTA (Sigma), and heated for 10 min at 95 °C. The pH was neutralized by adding 40 mM Tris-HCL (Sigma). Lysed cells were used as template for real-time qPCR, using 250 nM probe and 300 nM primers for exon1 (StepOne Plus system, Applied Biosystems) with TaqMan gene expression Master Mix (Applied Biosystems). Relative quantifications were performed using the comparative CT method (ΔCT) with *Gapdh* (Mm99999915_g1, genomic DNA compatible) (Applied Biosystems) as the reference gene. The average ΔCT for Cre-negative cells was used as the reference value for relative quantification.

**ELISA**. A stripe of tissue in the middle of the ear was homogenized in modified RIPA buffer as described[51] using a plastic pestle followed by centrifugation for 5 min at 5000×g. Supernatant was collected and total protein concentration was measured by BCA protein assay kit (Thermo Fisher Scientific). Three micrograms of total protein was used for ELISA assay using mouse VEGF-C kit (CUSABIO).

**Statistical analysis**. GraphPad Prism was used for graphic presentation and statistical analysis of the data. Data between two groups were compared with unpaired two-tailed Student's *t* test or Fisher's exact test, assuming equal variance. Comparison between multiple groups (Fig. 7d) was done using one-way ANOVA with Tukey's post hoc test. Differences were considered statistically significant when $P < 0.05$. The experiments were not randomized. No blinding was done in the analysis and quantifications.

**Data availability**. All relevant data are available from the corresponding author upon request.

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

## Acknowledgements

We thank Erwin Wagner (CNIO, Madrid) for the *Vegfr2*^flox mice, BioVis facility at Uppsala University for help with flow cytometry, Sofie Wagenius and Henrik Ortsäter (Uppsala University) for technical assistance, Katie Bentley (Uppsala University and Boston University) for discussion, and Ingvar Ferby (Uppsala University) for critical comments on the manuscript. This study was supported by the Swedish Research Council (542-2014-3535), the European Research Council (ERC-2014-CoG-646849), Knut and Alice Wallenberg Foundation (2015.0030), and the Kjell and Märta Beijer Foundation (to T.M.). M.F. was supported by postdoctoral fellowships from Lymphatic Education & Research Network and GA Johansson's Foundation.

## Author contributions

Y.Z., M.H.U., L.S., M.F., and I.M.-C. designed and performed experiments and analyzed data (Y.Z.: analysis of post-natal in vivo phenotypes and in vitro experiments, with contributions from M.F. and I.M.-C., M.H.U.: flow cytometry experiments and conceptual ideas, L.S.: analysis of embryonic phenotypes, with contribution from I.M.C.); K. A. provided essential tools and advice; T.M. directed the study and wrote the manuscript; all authors discussed the results and commented on the manuscript.

## Additional information

**Competing interests:** The authors declare no competing interests.

