## [Peer Review File(PDF 406 kb) · Nature Communications]

Reviewers' comments:

Reviewer #1 (Remarks to the Author):

The study by Zhang et al demonstrates an unexpected non-autonomous role for Vegfr3 during the formation of the lymphatic vascular network in mice. Using a series of careful inducible lymphatic specific KO experiments, the authors show that loss of function for Vegfr3 leads to increased vessel network branching rather than decreased lymphangiogenesis. Upon dissection of the causes of the phenotype, they show that in fact this occurs due to mosaic loss of Vegfr3. In this context, the non-deleted lymphatic endothelial cells show increased proliferation, increased anastomosis and form excessive network branches. This is not due to increased Vegfr2 activity (at least autonomously) as double loss of Vegfr2 and Vegfr3 exacerbates the phenotype. Mechanistically, the authors suggest that this occurs due to decreased Dll4 expression, which leads to increased proliferation in wt LECs in a LEC-LEC contact dependent manner. This later mechanism is supported by nicely performed in vitro studies.

Overall, this is a very well performed study that identifies both an interesting mechanism of vascular development and may explain some of the inconsistencies in the existing literature. It will be of interest to the angiogenesis and vascular biology field and is suitable for publication in Nature communications pending several revisions indicated below.

1. The reduction in Dll4 expression in Vegfr3 deficient LECs in vivo is only confirmed by QPCR from FAC sorted populations. Can this be further validated by immuno staining of tissues which display this hyperbranching phenotype? Or using an independent method?

2. Abnormal interconnections/anastomosis are an intriguing element of this phenotype but are only suggested because of decreased blunt ended vessels in the network. Can these ectopic anastomosis events be shown to occur upon mosaic loss of Vegfr3 using any additional system? Further to this, the authors suggest that it is mosaically reduced Dll4 expression that causes this phenotype. Is loss of Dll4 mosaically in any in vitro, in vivo system (or even a computational model) sufficient to phenocopy this observed increase in vessel interconnections? This last point is key as the current data does not bring the Dll4 function (in mosaic experiments) back to the abnormal EC behaviours that underly network hyperbranching in vivo.

3. The analysis of Vegfc levels in mutant ears by ELISA suggests that an increase in Vegfc levels is not sufficient to explain the phenotype. This one assay is useful but not rigorous. This reviewer does appreciate that quantifying Vegf levels in vivo is not at all easy, however the addition of a further readout or additional evidence that Vegfc is not increased would improve the confidence in this claim.

4. The in vitro experiments would be further strengthened by analysis of the dynamics of EC-EC interactions in mosaic co-cultures using time-lapse. This relates to point #2 above. Analysis of dynamic cell behaviours in culture in mosaic KO/siRNA conditions may lend additional support to the claims that phenotypes include ectopic anastomosis events or increased formation of connections.

5. The authors generate Vegfr3, Vegfr2 double KO LECs using the inducible system. They still see an increase in network anastomosis/hyperbranching. The conclusion is that Vegfr2 is therefore not responsible "autonomously" downstream of Vegfr3 but this is skipped over rather quickly in the manuscript. Presumably, the increased connections were formed by the non-deleted cells in double KOs (?) and the non-autonomous activation of Vegfr2 signalling (as well as Vegfr3) is a component of the mechanism? This should be discussed.

6. The study examines only one pathway in one tissue. The claims that this in some way generally

raises issues for future use of gene therapy are a little over the top.

Reviewer #2 (Remarks to the Author):

The manuscript by Zhang and colleagues is a well-written and highly interesting report of the unexpected mechanisms through which lymphatic vessel hyperplasia arises in VEGFR3 conditional deletion models. A similar hyperplastic phenotype has been previously described in an inducible model of blood vascular VEGFR3 deletion (Tammela et al, 2011; Heinolainen et al, 2017). However, the authors show that in the context of post-natal lymphangiogenesis this phenotype is not dependent upon a role of VEGFR3 as a negative regulator of the VEGF/VEGFR2 pathway.

The authors carefully and very convincingly demonstrate that lymphatic conditional deletion of VEGFR3 drives non cell autonomous proliferation of WT cells, therefore leading to the continued maintenance a pool of VEGFR3 positive cells in the vascular plexus, despite repeated rounds of induced deletion.

The authors convincingly suggest that direct contact interactions between VEGFR3- and VEGFR3+ positive cells are necessary for the increased proliferation observed in the latter cells. This non-cell autonomous increase in proliferation is Notch dependent.

In addition, the authors demonstrate the importance of VEGFR3 in tip cell function during lymphangiogenesis, consistent with its role during blood vessel angiogenesis.

This work has profound implications for the field and might lead to important reconsideration of the etiology of reported vascular (and non vascular) phenotypes in both inducible knockout models and pathological contexts.

I have only few minor comments about the manuscript, which I believe can be addressed in a reasonable time frame.

- Given the emphasis placed on the importance of Notch signalling as a mediator of this non cell-autonomous mechanism, I would suggest including an assessment of cell proliferation and direct contact proportions in co-culture (similar to what is presented in Figure 4) in a context of Notch signalling inhibition.

- Could the authors comment on the phenotypic differences between the embryonic vascular hypoplastic phenotype and the postnatal vascular hyperplasia? The vascular context appears to be a obvious difference, but it appears surprising that the vascular phenotype would be so diametrically opposed, given the large number of signalling pathways conserved between embryonic and postnatal lymphangiogenesis.

- I suggest that the authors include an experiment similar to that presented in Figure 4 but in a sub-confluent culture context. It would be interesting to further confirm that the increase in proliferation in WT cells occurs predominantly in cells directly contacting VEGFR- cells versus isolated cells.

- Have the authors investigated whether the blood vascular hyperplastic phenotype observed in *Pdgfb-iCreERT2; Vegfr3flox/flox* retinas can also be attributed in part to the non-cell autonomous mechanism described here?

- Do *Vegfr3flox/flox; Prox1-CreERT2* have any blood vascular phenotypes?

Reviewer #3 (Remarks to the Author):

This study by Zhang et al. reports a novel mechanism by which tissue expansion, in this case lymphatic vessels, can be driven by heterogeneity in growth factor signaling. Vegf-c and its receptor Vegfr-3 are well known lymphangiogenesis regulators. Previous studies using germ-line knockout models amply documented absolute requirement of this signaling pathway for the formation of lymphatic capillary network. Yet, here the authors report that conditional inactivation of Vegfr-3 can, in some situations, result in the expansion and not regression of lymphatic vessels.

First, Zhang et al. show that, as expected deletion of Vegfr-3 in embryonic LECs, leads to lymphatic vascular hypoplasia. However, when Vegfr-3 inactivation is initiated postnatally, denser lymphatic vasculature is observed in mouse ears – a tissue, in which lymphatic vessels undergo growth and remodeling at this stage. Further analysis of LEC proliferation shows that non-deleted LECs proliferate at much higher rate, thus accounting for the denser network at the endpoint, and co-culture experiments with mixed Vegfr-3 knockout and Vegfr-3 wild type LEC populations further confirm this phenotype. As proliferating WT LECs appear to be more frequently associated with KO LECs, the authors postulate the involvement of direct cell-cell contact and Notch signaling and show that DAPT treatment increases proliferation of VEGFR-3+ but not VEGFR-3 – LECs treated with low dose VEGF-C in vitro. Based on this and reduced expression of Notch ligand Dll4 and Hey1, a target of Notch signaling, in sorted mutant LECs, they conclude that loss of Dll4 and reduced Notch activation drive overproliferation of wild type LECs surrounded by mutant cells.

It is a carefully performed, important study, potentially; such mechanisms could drive e.g. development of vascular malformations caused by somatic mutations. There is however some important questions that still need to be addressed.

Major comments

1. The assumption is that in vivo Vegf-c/Vegfr-3 signaling drives overproliferation. While it is highly likely, it is not proven – please confirm that the hyperplastic phenotype can be rescued by global Vegfr-3 inhibition, using e.g. VEGFR-3 trap or blocking antibodies.
2. It will be important to discuss and present some additional data that clarify how general the proposed mechanism is. A priori, the model – overproliferation of non-deleted Vegfr-3+ LECs in the presence of deleted Vegfr-3- LECs - is very general and should be observed in every tissue with active Vegf-c-dependent lymphangiogenesis. Is this the case? Can the authors observe this phenotype upon mosaic Vegfr-3 inactivation during embryogenesis? Do the authors see lymphatic overgrowth in other tissues, such as gut or heart?
3. The authors show that Vegf-c protein concentration is not changed in the lysates from the control or mutant Vegfr-3 ears. This analysis is however rather crude and unlikely to catch changes in local Vegf-c concentrations or activation status, which could drive local non-deleted LEC proliferation. As Vegf-c is also known to be produced by e.g. endothelial cells, to support these data, please confirm that the expression of Vegf-c and another Vegfr-3 ligand Vegf-d in deleted vs. non-deleted LECs, dermal BECs and other stromal cells is not modified.
4. The authors conclude that VEGFR-3 KO LECs induce proliferation of VEGFR-3 WT LECs based on the probability calculations. Please also confirm experimentally that a co-culture of VEGFR-3 KO LECs and VEGFR-3 WT LECs without direct contact, e.g., on Transwell, does not affect VEGFR-3 WT LEC proliferation.
5. Additional data in support of the involvement of Notch signaling need to be provided:
 - DAPT has also a number of side effects not related to Notch inhibition, please confirm that increased proliferation is observed in Dll4 knockout LECs in vitro.

- How specific is the effect of VEGFR-3 –Notch cross talk on LEC proliferation? For example, do Notch-inhibited LECs also proliferate more in response to bFGF?
- Decreased expression of Hey1 in sorted mutant LECs is encouraging, however analysis of few additional Notch target genes, such as Nrarp and Hey2 would be helpful to show that indeed the whole pathway is shut down.
- Please provide the analysis of the Dll4 expression in the control and Vegfr-3 mutant ears. If Vegfr-3 signaling induces Dll4 in WT LECs, it should be especially high in non-deleted Vegfr-3+ tomato+ cells in Vegfr-3 mutant model, is this the case? A qPCR analysis similar to the one in Figure 5c and staining for Dll4 could be performed.

6. The role of Notch signaling in lymphatic vasculature is more controversial than discussed in the manuscript. While the study of Zheng et al., (Blood 2011) indeed shows that administration of Dll4-Fc induces lymphatic vessel hypersprouting, Niessen et al. (Blood, 2011) show that Dll4 and Notch1 blocking antibodies restrict growth and sprouting of postnatal dermal lymphatics as well as lymphangiogenesis during wound healing. These two groups used different reagents, time lines etc, nevertheless the role of Notch in lymphatics is still not clear to date. Therefore, the strongest evidence for the proposed model would be a genetic loss of Dll4 in LECs during the same time frame as Vegfr-3 inactivation in pups - such mice should have the same hyperproliferation and increased anastomosis phenotype.

Response to the Reviewers

NCOMMS-17-16093-T

We thank the Reviewers for their constructive comments that helped to improve the study. We have revised the manuscript and added several new experiments. In particular, our new data show that:

- DLL4 protein is downregulated in *Vegfr3* deleted dermal LECs in vivo, consistent with the reduction in mRNA expression presented in the original manuscript
- Mosaic loss of DLL4 promotes proliferation of neighboring WT LECs in vitro, thus phenocopying the effect of mosaic loss of VEGFR3
- VEGF-C, through signaling via both VEGFR2 and VEGFR3 in the non-targeted LECs, drives lymphatic vascular hyperplasia in the *Vegfr3^{fllox/fllox};R26-mTmG;Prox1-CreER^{T2}* mutant skin
- Embryonic requirement of VEGFR3 for LEC survival and proliferation is conserved in early postnatal vasculature of the ear
- Lymphatic hyperplasia induced by mosaic loss of VEGFR3 is not limited to skin, but can be observed also in the intestinal wall

To comply with the formatting guidelines, we have added introduction and discussion in the revised manuscript. In addition, the number of figures was increased to 7 (previously 5) after new data was added. A manuscript file with main changes highlighted in red is provided.

Specific points were addressed as follows (in blue, new figure panels indicated in bold):

Reviewer #1

The study by Zhang et al demonstrates an unexpected non-autonomous role for Vegfr3 during the formation of the lymphatic vascular network in mice. Using a series of careful inducible lymphatic specific KO experiments, the authors show that loss of function for Vegfr3 leads to increased vessel network branching rather than decreased lymphangiogenesis. Upon dissection of the causes of the phenotype, they show that in fact this occurs due to mosaic loss of Vegfr3. In this context, the non-deleted lymphatic endothelial cells show increased proliferation, increased anastomosis and form excessive network branches. This is not due to increased Vegfr2 activity (at least autonomously) as double loss of Vegfr2 and Vegfr3 exacerbates the phenotype. Mechanistically, the authors suggest that this occurs due to decreased Dll4 expression, which leads to increased proliferation in wt LECs in a LEC-LEC contact dependent manner. This later mechanism is supported by nicely performed in vitro studies.

Overall, this is a very well performed study that identifies both an interesting mechanism of vascular development and may explain some of the inconsistencies in the existing literature. It will be of interest to the angiogenesis and vascular biology field and is suitable for publication in Nature communications pending several revisions indicated below.

1. The reduction in Dll4 expression in Vegfr3 deficient LECs in vivo is only confirmed by QPCR from FAC sorted populations. Can this be further validated by immuno staining of tissues which display this hyperbranching phenotype? Or using an independent method?

Response: As suggested, we have performed immunostaining and show that DLL4 protein is downregulated in the *Vegfr3^{fllox/fllox};R26-mTmG;Prox1-CreER^{T2}* mutant vasculature specifically in GFP+ (i.e. VEGFR3⁺) LECs in comparison to non-targeted GFP- (i.e. VEGFR3⁺) LECs (**Fig. 6c**).

2. Abnormal interconnections/anastomosis are an intriguing element of this phenotype but are only suggested because of decreased blunt ended vessels in the network. Can these ectopic anastomosis events be shown to occur upon mosaic loss of Vegfr3 using any additional system?

Response: We are not aware of any suitable in vitro system that could be utilized to investigate LEC anastomosis process. However, to increase confidence in our conclusion that anastomosis is a critical element of the phenotype, we provide new data showing that excessive anastomosis is an active process driven by non-targeted VEGFR3⁺ cells when in contact with VEGFR3 deleted cells. Inhibition of VEGFR3 in the non-targeted cells using the kinase inhibitor MAZ51 significantly reduced vessel anastomosis and restored the number of blunt-ended vessels in the central region of the ear in the *Vegfr3^{fllox/fllox};R26-mTmG;Prox1-CreER^{T2}* mutants (**Fig. 3f-h**).

Further to this, the authors suggest that it is mosaically reduced Dll4 expression that causes this phenotype. Is loss of Dll4 mosaically in any in vitro, in vivo system (or even a computational model) sufficient to phenocopy this observed increase in vessel interconnections? This last point is key as the current data does not bring the Dll4 function (in mosaic experiments) back to the abnormal EC behaviours that underly network hyperbranching in vivo.

Response: As discussed above, we do not have currently any in vitro systems that would allow us to investigate LEC anastomosis process. Generation of genetic mutants is not feasible in a reasonable time frame. As we previously demonstrated, in addition to anastomosis, VEGFR3-driven induction of proliferation in the neighbouring VEGFR3⁺ cells is a key mechanism underlying the hyperplastic phenotype in the mutant vasculature. We have therefore addressed the important question of whether mosaic loss of DLL4 can recapitulate the effect of mosaic loss of VEGFR3 by assessing the effect on proliferation.

For this purpose, we performed co-culture experiments using LECs treated with control siRNA and siRNA against DLL4. As expected, and consistent with our finding that DAPT treatment increases LEC proliferation (Fig. 7a), DLL4 siRNA treated LECs showed increased proliferation in comparison to control siRNA treated cells when cultured alone (Fig. 7d). Importantly, increasing the proportion of siDLL4 cells in the co-cultures increased proliferation in siCTRL LECs (Fig. 7d). We thus conclude that loss of DLL4 mosaically is sufficient to phenocopy the effect of mosaic loss of VEGFR3 on the behavior of neighboring cells, which contributes to vessel hyperplasia in vivo.

3. The analysis of Vegfc levels in mutant ears by ELISA suggests that an increase in Vegfc levels is not sufficient to explain the phenotype. This one assay is useful but not rigorous. This reviewer does appreciate that quantifying Vegf levels in vivo is not at all easy, however the addition of a further readout or additional evidence that Vegfc is not increased would improve the confidence in this claim.

Response: In addition to our previous analysis of ear lysates for VEGF-C protein, we now provide evidence that *Vegfc* mRNA levels are not increased in the mutant skin. We analyzed *Vegfc* (and *Vegfd*) mRNA expression in whole ears, BECs and targeted and non-targeted LECs (Supplementary Fig. 7a, b). Of note, expression of *Vegfc* and *Vegfd* was very low or absent in LECs of both controls and mutants. In addition, we show that macrophages, which are major producers of VEGF-C and VEGFD in vivo^{1,2}, are not increased in the mutant skin (Supplementary Fig. 7c). We have not been able to successfully stain for VEGF-C protein in whole-mount ears in vivo, neither have we seen such immunostainings in the literature (except in situations of transgenic overexpression of VEGF-C).

4. The in vitro experiments would be further strengthened by analysis of the dynamics of EC-EC interactions in mosaic co-cultures using time-lapse. This relates to point #2 above. Analysis of dynamic cell behaviours in culture in mosaic KO/siRNA conditions may lend additional support to the claims that phenotypes include ectopic anastomosis events or increased formation of connections.

Response: As suggested, we have attempted to investigate the dynamics of cell-cell interactions in co-cultures of VEGFR3⁺ and VEGFR3⁻ LECs *in vitro*. However, LECs cultured in 2D do not allow studying ‘anastomosis events’ because unlike BECs that are able to move individually and extend ‘tip cell-like’ protrusion towards other BECs, LECs form tight clusters that move collectively even when sparsely plated (³ and data not shown). Importantly, we found that under subconfluent conditions when proliferation rate is high, LECs do not respond to Notch inhibition by further increasing their proliferation (see response to Reviewer 2’s question 3; **Figure 2 for the Reviewers**). On the other hand, under confluent conditions when Notch-dependent regulation of LEC proliferation is observed (Fig. 7a, b), we found that LECs show minimal cell movements and rearrangements (**Fig. 1 for the Reviewers**). We performed Incucyte time-lapse imaging to allow tracking the movement of individual cells in primary cultures of dermal LEC isolated from *Vegfr3^{fllox/fllox};R26-mTmG;Prox1-CreER^{T2}* mice after 4-OHT mediated *Vegfr3* deletion (as shown in Fig. 5b). Measurement of the total distance covered over 24 h period showed no differences between WT cells with or without a direct contact

with KO cells in comparison the KO cells (**Fig. 1 for the Reviewers**).

Fig. 1 for the Reviewers. Tracking of movement of individual LECs in cultures of primary mouse LECs from *Vegfr3^{flox/flox};R26-mTmG;Prox1-CreER^{T2}* mice. KO cells were identified based on GFP expression after in vitro 4-OHT mediated gene deletion. Total distance covered (left) and cell movement tracks (right) during a period of 24 h are shown. No differences were observed in the movement of WT cells with (n=10) or without (n=10) a direct contact with KO cells in comparison the KO cells (n=10).

While the anastomosis process is very interesting, and we have been able to provide additional data showing that excessive anastomosis is actively driven by non-targeted VEGFR3⁺ cells when in the context of VEGFR3⁻ cells in vivo (**Fig. 3f, g**), we have avoided over interpreting this data as explaining the lymphatic hyperplasia in the *Vegfr3* mutants. It is likely that several processes (vessel anastomosis, cell proliferation and migration) contribute to the phenotype. We assessed the effect of *Vegfr3* deleted LECs on the behavior of neighboring LECs by focusing on one aspect namely cell proliferation, which we have been able to investigate both in vivo (Fig. 3) and in vitro (Fig. 5).

5. The authors generate *Vegfr3*, *Vegfr2* double KO LECs using the inducible system. They still see an increase in network anastomosis/hyperbranching. The conclusion is that *Vegfr2* is therefore not responsible “autonomously” downstream of *Vegfr3* but this is skipped over rather quickly in the manuscript. Presumably, the increased connections were formed by the non-deleted cells in double KOs (?) and the non-autonomous activation of *Vegfr2* signalling (as well as *Vegfr3*) is a component

of the mechanism? This should be discussed.

Response: We provide new data showing that abnormal vessel connections in the double KO mice are indeed formed by non-targeted LECs (**Supplementary Fig. 4g**). To further assess the contribution of VEGF-C signaling to the phenotype, due to its ability to stimulate non-targeted LECs, we inhibited all VEGF-C signaling in the *Vegfr3^{fllox/fllox};R26-mTmG;Prox1-CreER^{T2}* mutant and Cre⁻ littermate control mice by administering adeno-associated vectors (AAVs) encoding soluble VEGF-C-trap (AAV-Vegfr3-Ig)⁴. The relative contribution of the two VEGF receptors in non-targeted LECs was further investigated by treatment with the VEGFR2 blocking antibody DC101 or the VEGFR3 kinase inhibitor MAZ51.

To overcome the requirement of VEGF-C signaling for LEC survival and proliferation at early postnatal stages⁵, while achieving efficient inhibition during the critical period for phenotype to develop (Fig. 2c), we administered AAV-VEGFR3-Ig by intraperitoneal injection at P10. As expected, control mice expressing VEGFR3-Ig showed decreased growth of lymphatic vasculature in the ear skin (**Fig. 3c, d**). In the *Vegfr3^{fllox/fllox};R26-mTmG;Prox1-CreER^{T2}* mutants, hyperbranching phenotype was completely rescued and the vasculature showed a similar reduction in growth as compared to the control (**Fig. 3c, d**). Selective VEGFR3 inhibition from P7 did not affect the ear vasculature in the control mice when analysed at P21, but partially rescued hyperbranching in 2 out of 4 mutants (**Fig. 3e**). Interestingly, VEGFR3 inhibition significantly inhibited vessel anastomosis and restored the number of blunt-ended vessels in the central region of the ear (**Fig. 3f-h**). In contrast, blocking VEGFR2 signaling inhibited *Vegfr3* loss induced hyperbranching but the number of blunt-ended vessels was not increased (**Fig. 3g, h**).

Together, these data show that VEGF-C, through signaling via both VEGFR2 and VEGFR3 in the non-targeted LECs, drives lymphatic vascular hyperplasia in the *Vegfr3^{fllox/fllox};R26-mTmG;Prox1-CreER^{T2}* mutant skin.

We have also added the following text in the discussion:

‘These data suggest specific and synergistic functions of VEGFR2 and VEGFR3 in driving abnormal VEGF-C-driven behavior of non-targeted VEGFR3⁺ cells when in contact with VEGFR3⁻ cells. These results should, however, be interpreted with some caution since it is not possible to ensure complete or equal inhibition of the two receptors with the inhibitors. That MAZ51 did not have an apparent effect on wild type vasculature at P21 indeed suggests that the inhibitor does not fully inhibit VEGFR3 activity. Another consideration is that while MAZ51 targets only the non-recombined VEGFR3⁺ cells in the mutant skin, VEGF-C-trap and DC101 target both the recombined VEGFR3⁻ and non-recombined VEGFR3⁺ LECs by inhibiting VEGFR2 signaling. Although LEC-specific genetic deletion of *Vegfr2* alone or in combination with *Vegfr3* did not indicate a function for VEGFR2 in the LECs, DC101 slightly reduced lymphatic growth in wild type mice. It is thus not possible to exclude a minor role for VEGFR2 in lymphatic development or in the VEGFR3 deleted

cells in the mutant, which may be masked e.g. due to slow kinetics of VEGFR2 depletion in the genetic mutants.’

6. *The study examines only one pathway in one tissue. The claims that this in some way generally raises issues for future use of gene therapy are a little over the top.*

Response: We understand the Reviewer’s concern about the generality of our findings. We now show that lymphatic hyperplasia induced by mosaic loss of VEGFR3 is not limited to skin but can be observed also in the intestinal wall (**Supplementary Fig. 3b, c**). However, we argue that the implications of our findings are not limited to the VEGFR3 signaling pathway or the lymphatic vasculature. The phenomenon we have described here by which a mosaic loss/inhibition of a growth promoting pathway can unexpectedly promote tissue growth through non-cell-autonomous effects can be envisaged to apply to any tissue and growth promoting signaling pathway. Recently, such concept is emerging in cancer, where interactions between different subclones of tumor cells can influence tumor progression⁶. It is also recognized that this clonal cooperation can have a profound effect on therapeutic outcomes, and identification of non-cell-autonomous driver subclones that promote tumour growth by influencing nearby populations is critical⁶. Since a major problem of therapeutic approaches such as siRNA nanovectors is an incomplete delivery to the target cell population, leading to generation of subclones with heterogeneous expression of the target gene, it becomes important to consider such non-cell-autonomous effects. We have added discussion on this topic (**page 19**).

Reviewer #2

The manuscript by Zhang and colleagues is a well-written and highly interesting report of the unexpected mechanisms through which lymphatic vessel hyperplasia arises in VEGFR3 conditional deletion models. A similar hyperplastic phenotype has been previously described in an inducible model of blood vascular VEGFR3 deletion (Tammela et al, 2011; Heinolainen et al, 2017). However, the authors show that in the context of post-natal lymphangiogenesis this phenotype is not dependent upon a role of VEGFR3 as a negative regulator of the VEGF/VEGFR2 pathway.

The authors carefully and very convincingly demonstrate that lymphatic conditional deletion of VEGFR3 drives non cell autonomous proliferation of WT cells, therefore leading to the continued maintenance a pool of VEGFR3 positive cells in the vascular plexus, despite repeated rounds of induced deletion.

The authors convincingly suggest that direct contact interactions between VEGFR3- and VEGFR3+

positive cells are necessary for the increased proliferation observed in the latter cells. This non-cell autonomous increase in proliferation is Notch dependent.

In addition, the authors demonstrate the importance of VEGFR3 in tip cell function during lymphangiogenesis, consistent with its role during blood vessel angiogenesis.

This work has profound implications for the field and might lead to important reconsideration of the etiology of reported vascular (and non vascular) phenotypes in both inducible knockout models and pathological contexts.

I have only few minor comments about the manuscript, which I believe can be addressed in a reasonable time frame.

1- Given the emphasis placed on the importance of Notch signalling as a mediator of this non cell-autonomous mechanism, I would suggest including an assessment of cell proliferation and direct contact proportions in co-culture (similar to what is presented in Figure 4) in a context of Notch signalling inhibition.

Response: As suggested, we performed co-culture experiments using LECs treated with control siRNA and siRNA against DLL4. As expected, and consistent with our finding that DAPT treatment increases LEC proliferation (Fig. 7a, b), DLL4 siRNA treated LECs showed increased proliferation in comparison to control siRNA treated cells when cultured alone (**Fig. 7d**). Importantly, increasing the proportion of siDLL4 cells in the co-cultures increased proliferation in siCTRL LECs (**Fig. 7d**).

2- Could the authors comment on the phenotypic differences between the embryonic vascular hypoplastic phenotype and the postnatal vascular hyperplasia? The vascular context appears to be a obvious difference, but it appears surprising that the vascular phenotype would be so diametrically opposed, given the large number of signalling pathways conserved between embryonic and postnatal lymphangiogenesis.

Response: The Reviewer asks a valid question. Lymphatic hyperplasia caused by early postnatal *Vegfr3* deletion was indeed unexpected considering the previously established pro-lymphangiogenic role of VEGF-C signaling both during embryogenesis (^{7,8}; Fig. 1 this study) and early postnatal development^{5,9}.

To better understand the apparent controversy, we attempted to carefully assess LEC proliferation and survival, known to depend on VEGF-C/VEGFR3 signaling during embryogenesis, in early postnatal development. Continuous induction of *Vegfr3* deletion in a

new pool of non-recombined LECs over a period of several days (Fig. 4) did not allow following the destiny of *Vegfr3* deleted cells and assessing the effect on LEC survival. For this purpose, we performed a pulsed induction of *Vegfr3* deletion by a single dose of 4-OHT that was administered during an active growth phase between P7-P10 (**Supplementary Fig. 6a**). An overall reduction of GFP⁺ cells in the mutant in comparison to heterozygous vasculature was observed 2 days after induction at P7 or P10 (**Supplementary Fig. 6b, c**). This reduction could be referred to a selective reduction in GFP⁺ cells that had lost Tomato (**Supplementary Fig. 6b, d, e**), suggesting decreased survival of *Vegfr3* deficient LECs (Fig. 4c) at this stage. *Vegfr3* deficient LECs also showed a significantly lower EdU incorporation rate compared to LECs in control embryos (**Supplementary Fig. 6f**), similar to observed in E15 skin (**Supplementary Fig. 6g**). The requirement of VEGFR3 for LEC survival and proliferation selectively at the early postnatal stages was corroborated by VEGF-C-trap-induced regression of dermal lymphatic vasculature in the ear when administered at P7 (**Supplementary Fig. 6h**), but not after P14⁵.

Taken together, these results demonstrate that during the initial growth of lymphatic vascular plexus VEGFR3 is required for LEC survival and proliferation, both in the dorsal skin of embryos (Fig. 1) and the ear skin of early postnatal mice (**Supplementary Fig. 6**). Embryonic requirement of VEGFR3 for LEC survival and proliferation is thus conserved in early postnatal vasculature of the ear.

3- I suggest that the authors include an experiment similar to that presented in Figure 4 but in a sub-confluent culture context. It would be interesting to further confirm that the increase in proliferation in WT cells occurs predominantly in cells directly contacting VEGFR- cells versus isolated cells.

Response: We agree with the Reviewer that the suggested experiment would be potentially interesting. However, we find that under sub-confluent conditions LEC proliferation is high overall and, unlike under confluent conditions (Fig. 7a, b), we cannot observe an increase in proliferation upon Notch inhibition (**Fig. 2 for the Reviewers**). Since VEGFR3 deleted LECs exert their effect on proliferation through downregulation of Notch signaling in the neighboring cells, it would thus not be possible to assess this mechanism under sub-confluent conditions.

Fig. 2 for the Reviewers. DAPT induced effect on the proliferation of LECs cultured under subconfluent conditions (3.2×10^4 cells/cm²). Tile scan immunofluorescence images showing staining for EdU⁺ nuclei after 3 h of incorporation (representative of n=3 biological replicates). Scale bars: 250 μ m.

4- Have the authors investigated whether the blood vascular hyperplastic phenotype observed in *Pdgfb-iCreERT2; Vegfr3flox/flox* retinas can also be attributed in part to the non-cell autonomous mechanism described here?

Response: In BECs VEGFR3 deletion has been shown to lead to upregulation of the major angiogenic VEGFR2 receptor, both in vitro and in vivo^{10,11}. As shown in our study (Supplementary Fig. 4d-f), this is not the case in LECs. The differential regulation of VEGFR2 by VEGFR3 in the two cell types likely accounts for the different mechanisms driving the phenotype. We have now clarified this in the discussion. Whether a non-cell autonomous mechanism additionally contributes to retinal vascular hyperplasia observed upon loss of *Vegfr3* warrants a separate study where this question can be studied in sufficient detail.

5- Do *Vegfr3flox/flox; Prox1-CreERT2* have any blood vascular phenotypes?

Response: The *Prox1-CreER²* line targets specifically the dermal lymphatic vasculature in the postnatal ear (^{12,13} and this study). To exclude a non-cell autonomous effect of LEC-specific *Vegfr3* deletion on the blood vasculature, we provide images of whole-mount stained ears showing normal blood vasculature in the mutant in comparison to the control (**Supplementary Fig. 3a**).

Reviewer #3

This study by Zhang et al. reports a novel mechanism by which tissue expansion, in this case lymphatic vessels, can be driven by heterogeneity in growth factor signaling. Vegf-c and its receptor Vegfr-3 are well known lymphangiogenesis regulators. Previous studies using germ-line knockout models amply documented absolute requirement of this signaling pathway for the formation of lymphatic capillary network. Yet, here the authors report that conditional inactivation of Vegfr-3 can, in some situations, result in the expansion and not regression of lymphatic vessels.

First, Zhang et al. show that, as expected deletion of Vegfr-3 in embryonic LECs, leads to lymphatic vascular hypoplasia. However, when Vegfr-3 inactivation is initiated postnatally, denser lymphatic vasculature is observed in mouse ears – a tissue, in which lymphatic vessels undergo growth and remodeling at this stage. Further analysis of LEC proliferation shows that non-deleted LECs proliferate at much higher rate, thus accounting for the denser network at the endpoint, and co-culture experiments with mixed Vegfr-3 knockout and Vegfr-3 wild type LEC populations further confirm this phenotype. As proliferating WT LECs appear to be more frequently associated with KO LECs, the authors postulate the involvement of direct cell-cell contact and Notch signaling and show that DAPT treatment increases proliferation of VEGFR-3+ but not VEGFR-3 – LECs treated with low dose VEGF-C in vitro. Based on this and reduced expression of Notch ligand Dll4 and Hey1, a target of Notch signaling, in sorted mutant LECs, they conclude that loss of Dll4 and reduced Notch activation drive overproliferation of wild type LECs surrounded by mutant cells.

It is a carefully performed, important study, potentially; such mechanisms could drive e.g. development of vascular malformations caused by somatic mutations. There is however some important questions that still need to be addressed.

Major comments

1. The assumption is that in vivo Vegf-c/Vegfr-3 signaling drives overproliferation. While it is highly likely, it is not proven – please confirm that the hyperplastic phenotype can be rescued by global Vegfr-3 inhibition, using e.g. VEGFR-3 trap or blocking antibodies.

Response: To address the dependency of the hypersprouting phenotype on VEGF-C, we inhibited all VEGF-C signaling in the *Vegfr3^{fllox/fllox};R26-mTmG;Prox1-CreER^{T2}* mutant and Cre⁻ littermate control mice by administering adeno-associated vectors (AAVs) encoding soluble VEGF-C-trap (AAV-Vegfr3-Ig)⁴. The relative contribution of the two VEGF receptors in non-targeted LECs was further investigated by treatment with the VEGFR2 blocking antibody DC101 or the VEGFR3 kinase inhibitor MAZ51.

To overcome the requirement of VEGF-C signaling for LEC survival and proliferation at early postnatal stages⁵, while achieving efficient inhibition during the critical period for phenotype to develop (Fig. 2c), we administered AAV-VEGFR3-Ig by intraperitoneal injection at P10. As expected, control mice expressing VEGFR3-Ig showed decreased growth of lymphatic vasculature in the ear skin (Fig. 3c, d). In the *Vegfr3^{fllox/fllox};R26-mTmG;Prox1-CreER^{T2}* mutants, hyperbranching phenotype was completely rescued and the vasculature showed a similar reduction in growth as compared to the control (Fig. 3c, d). Selective VEGFR3 inhibition from P7 did not affect the ear vasculature in the control mice when analysed at P21, but partially rescued hyperbranching in 2 out of 4 mutants (Fig. 3e). Interestingly, VEGFR3 inhibition significantly inhibited vessel anastomosis and restored the number of blunt-ended vessels in the central region of the ear (Fig. 3f-h). In contrast, blocking VEGFR2 signaling inhibited *Vegfr3* loss induced hyperbranching but the number of blunt-ended vessels was not increased (Fig. 3g, h).

Together, these data show that VEGF-C, through signaling via both VEGFR2 and VEGFR3 in the non-targeted LECs, drives lymphatic vascular hyperplasia in the *Vegfr3^{fllox/fllox};R26-mTmG;Prox1-CreER^{T2}* mutant skin.

2. It will be important to discuss and present some additional data that clarify how general the proposed mechanism is. A priori, the model – overproliferation of non-deleted Vegfr-3+ LECs in the presence of deleted Vegfr-3- LECs - is very general and should be observed in every tissue with active Vegf-c-dependent lymphangiogenesis. Is this the case? Can the authors observe this phenotype upon mosaic Vegfr-3 inactivation during embryogenesis? Do the authors see lymphatic overgrowth in other tissues, such as gut or heart?

Response: As suggested by the Reviewer, we have analyzed lymphatic vasculature in the gut and observe a similar hyperbranching phenotype in the intestinal wall (Supplementary Fig. 3, c). Of note, whole-mount analysis of non-targeted and targeted LECs was not possible in this tissue due to efficient recombination and widespread GFP expression in the intestinal wall muscle layer in the *Prox1-CreER^{T2};R26-mTmG* mice.

We also analyzed the cardiac lymphatic vasculature in the mutant in comparison to control mice at 3 weeks of age, but could not obtain conclusive data. This was, at least partly, because we found that

the patterning of the cardiac vasculature shows inter-individual variation. Considering that the cardiac lymphatic vasculature forms during embryonic development¹⁴, our gene deletion and phenotype analysis during postnatal development may not be relevant. Careful analysis of different time points of cardiac development would be needed to obtain conclusive data, which we feel goes beyond the scope of this study.

Hyperbranching of dermal lymphatic vasculature was not observed upon LEC-specific deletion of *Vegfr3* during embryogenesis. Rather, as shown in Fig. 1b-g, this resulted in severe vessel hypoplasia, which is consistent with the critical function of VEGF-C/VEGFR3 signalling for LEC survival during embryonic development⁸. The function of VEGFR3 in LEC survival may thus dominate in this context, not allowing for long enough interactions between targeted and non-targeted cells to induce changes in Notch signaling.

The process by which VEGFR3⁻ LECs non-cell-autonomously regulate lymphatic vessel growth by inducing proliferation of non-targeted VEGFR3⁺ LECs through cell contact-dependent reduction in Notch signaling is thus unlikely to occur in every tissue with active VEGF-C-dependent lymphangiogenesis. It is possible that lymphatic vascular beds that form through a plexus intermediate via repeated cycles of vessel sprouting and anastomosis are particularly sensitive to aberrations in Notch signaling, due to the established function of this pathway in controlling similar processes in the blood vasculature^{15,16}. We have added discussion on this on **page 17-18**.

3. The authors show that Vegf-c protein concentration is not changed in the lysates from the control or mutant Vegfr-3 ears. This analysis is however rather crude and unlikely to catch changes in local Vegf-c concentrations or activation status, which could drive local non-deleted LEC proliferation. As Vegf-c is also known to be produced by e.g. endothelial cells, to support these data, please confirm that the expression of Vegf-c and another Vegfr-3 ligand Vegf-d in deleted vs. non-deleted LECs, dermal BECs and other stromal cells is not modified.

Response: In addition to our previous analysis of ear lysates for VEGF-C protein, we now provide evidence that *Vegfc* mRNA levels are not increased in the mutant skin. We analyzed *Vegfc* (and *Vegfd*) mRNA expression in whole ears, BECs and targeted and non-targeted LECs (**Supplementary Fig. 7a, b**). Of note, expression of *Vegfc* and *Vegfd* was very low or absent in LECs of both controls and mutants. In addition, we show that macrophages, which are major producers of VEGF-C and VEGFD in vivo^{1,2}, are not increased in the mutant skin (**Supplementary Fig. 7c**). We have not been able to successfully stain for VEGF-C protein in whole-mount ears in vivo, neither have we seen such immunostainings in the literature (except in situations of transgenic overexpression of VEGF-C).

4. The authors conclude that VEGFR-3 KO LECs induce proliferation of VEGFR-3 WT LECs based on the probability calculations. Please also confirm experimentally that a co-culture of VEGFR-3 KO

LECs and VEGFR-3 WT LECs without direct contact, e.g., on Transwell, does not affect VEGFR-3 WT LEC proliferation.

Response: We have performed Transwell co-culture experiment as suggested. Proliferation of siCTRL cells was not affected when co-cultured without a direct contact with siVEGFR3 cells seeded on a Transwell culture insert (**Fig. 5g**).

5. Additional data in support of the involvement of Notch signaling need to be provided:

- DAPT has also a number of side effects not related to Notch inhibition, please confirm that increased proliferation is observed in Dll4 knockout LECs in vitro.

Response: We analyzed LEC proliferation in LECs treated with control siRNA and siRNA against DLL4 or NOTCH1. As expected, NOTCH1 (**Fig. 7c**) or DLL4 siRNA treated LECs showed increased proliferation in comparison to control siRNA treated cells (**Fig. 7d**) when cultured alone. In addition, when co-cultured with control LECs cells, increasing the proportion of siDLL4 cells increased the proliferation in siCTRL LECs (**Fig. 7d**), thus phenocopying the effect of mosaic loss of VEGFR3.

- How specific is the effect of VEGFR-3 –Notch cross talk on LEC proliferation? For example, do Notch-inhibited LECs also proliferate more in response to bFGF?

Response: We now show that Notch inhibition can induce proliferation in LECs that are cultured in complete serum containing medium in the absence of VEGF-C (**Fig. 7c**). However, the effect is more pronounced when VEGF-C is present (**Fig. 7c**). Importantly, we show that in vivo the hyperbranching phenotype is driven by VEGF-C (**Fig. 3c-h**), indicating the relevance of VEGF-C for the studied process.

- Decreased expression of Hey1 in sorted mutant LECs is encouraging, however analysis of few additional Notch target genes, such as Nrarp and Hey2 would be helpful to show that indeed the whole pathway is shut down.

Response: As suggested, we analyzed the expression of *Nrarp* and *Hey2* in ECs sorted from ear skin. *Hey2* was not expressed at a detectable level in LECs in vivo (**Fig. 3a for the Reviewers**). *Nrarp* showed very low expression with high variation in wild type LECs and clear conclusions could thus not be drawn from this data (**Fig. 3b for the Reviewers**). However, we observed strong downregulation of *Efnb2*, a previously demonstrated target of DLL4-NOTCH1 signaling in the lymphatic vasculature¹⁷, in the non-targeted LECs (**Fig. 7e**). The demonstrated strong downregulation of DLL4 in VEGFR3 deficient cells, combined with the downregulation of *Hey1* and *Efnb2* in the

non-targeted cells, provides compelling evidence for downregulation of Notch pathway activity in the mutant vasculature in vivo.

Fig. 3 for the Reviewers. qRT-PCR analysis of *Hey2* (a) and *Nrarp* (b) expression in whole skin and FACS sorted BECs and LECs from P13-P14 *Cre*⁻ (*Ctrl*) and *Vegfr3*^{fllox/fllox}; *R26-mTmG*; *Prox1-CreER*^{T2} (*fl/fl*) mice. Results from non-targeted (Tomato⁺GFP⁺; R3+), and targeted (Tomato⁻GFP⁺; R3-) LECs are displayed separately. Horizontal lines represent mean relative expression (n=3-4 mice with n=2 technical replicates for each).

- Please provide the analysis of the *Dll4* expression in the control and *Vegfr-3* mutant ears. If *Vegfr-3* signaling induces *Dll4* in WT LECs, it should be especially high in non-deleted *Vegfr-3*+ tomato+ cells in *Vegfr-3* mutant model, is this the case? A qPCR analysis similar to the one in Figure 5c and staining for *Dll4* could be performed.

Response: To verify downregulation of *DLL4* in VEGFR3- LECs in the mutant skin also at the protein level, we performed whole-mount immunostaining. Consistent with the previously shown reduction in mRNA expression, we found that *DLL4* protein is downregulated in the *Vegfr3*^{fllox/fllox}; *R26-mTmG*; *Prox1-CreER*^{T2} mutant vasculature specifically in GFP+ (i.e. VEGFR3⁻) LECs in comparison to non-targeted GFP- (i.e. VEGFR3⁺) LECs (Fig. 7c).

6. The role of Notch signaling in lymphatic vasculature is more controversial than discussed in the manuscript. While the study of Zheng et al., (Blood 2011) indeed shows that administration of *Dll4*-Fc induces lymphatic vessel hypersprouting, Niessen et al. (Blood, 2011) show that *Dll4* and *Notch1*

blocking antibodies restrict growth and sprouting of postnatal dermal lymphatics as well as lymphangiogenesis during wound healing. These two groups used different reagents, time lines etc, nevertheless the role of Notch in lymphatics is still not clear to date. Therefore, the strongest evidence for the proposed model would be a genetic loss of Dll4 in LECs during the same time frame as Vegfr-3 inactivation in pups - such mice should have the same hyperproliferation and increased anastomosis phenotype.

Response: The Reviewer correctly highlights controversy around the role of Notch signalling in the lymphatic vasculature. With reformatting of the revised manuscript to comply with the formatting guidelines, a separate discussion paragraph was added. We were therefore able to extend discussion on the previously demonstrated context-dependent roles of Notch signalling in lymphatic development (**page 17-18**). Indeed, inhibition of DLL4-Notch1 signalling inhibits dermal lymphatic vessel growth and sprouting in neonatal mice¹⁷, but promotes vessel hyperplasia in adult mouse skin¹⁸. Of note, inhibition of VEGF-C signalling blocks lymphatic vessel growth and induces regression during embryonic and early postnatal development, but not after 2 weeks of age^{5,8}. Interestingly, our data demonstrate that the embryonic requirement of VEGFR3 is conserved in neonatal ear vasculature (**Supplementary Fig. 6**). Demonstration that VEGFR3 is an upstream regulator of DLL4 is thus consistent with the early dependence of lymphatic growth on DLL4-Notch1 signalling.

We have also addressed the Reviewer's question experimentally. Generation of genetic mutants was not feasible in a reasonable time frame. We therefore inhibited Notch signaling in early postnatal development of the ear using adeno-associated vectors (AAVs) encoding Dll4-Fc, which was previously shown to induce lymphatic vessel hypersprouting in the adult ear skin¹⁸. Although Dll4-Fc causes global rather than mosaic loss/inhibition of DLL4-Notch signaling, the effect mimics that observed in the *Vegfr3* deleted ears where both targeted and non-targeted cells show reduced Notch activity, caused by interaction with the VEGFR3⁺DLL4^{low} cells (Fig. 6d). We administered AAV-Dll4-Fc either into the ear or intraperitoneally at different stages of early postnatal development (P2, P10). In all cases, we observed a severe gross phenotype characterized by general growth retardation and bleeding in the intestine, and the experiments had to be terminated early. It was not possible to draw conclusions from the lymphatic vasculature, due to the young age of the mice, but we observed mild hyperbranching of the dermal blood vasculature (data not shown). This discouraged us from using the approach during early developmental stages, since any potential phenotype in the lymphatic vasculature could be caused indirectly by blood vessel defects.

Although we are not able to provide *in vivo* data from genetic Notch mouse models, our other data provide strong evidence for the contribution of DLL4-Notch signaling to lymphatic hyperplasia induced by mosaic loss of *Vegfr3* by showing that 1) DLL4 is downregulated at mRNA and protein levels in VEGFR3⁺ cells *in vivo* and *in vitro*, 2) Notch targets genes *Hey1* and *Efnb2* are also downregulated, and 3) mosaic loss of DLL4 is sufficient to phenocopy the effect of mosaic loss of VEGFR3 on the proliferation of neighboring cells in primary LEC cultures *in vitro*.

References

1. Schoppmann, S. F. *et al.* Tumor-associated macrophages express lymphatic endothelial growth factors and are related to peritumoral lymphangiogenesis. *Am. J. Pathol.* **161**, 947–956 (2002).
2. Harvey, N. L. & Gordon, E. J. Deciphering the roles of macrophages in developmental and inflammation stimulated lymphangiogenesis. *Vasc. Cell* **4**, 15 (2012).
3. Mäkinen, T. *et al.* Isolated lymphatic endothelial cells transduce growth, survival and migratory signals via the VEGF-C/D receptor VEGFR-3. *EMBO J.* **20**, 4762–4773 (2001).
4. Fang, S. *et al.* Critical requirement of VEGF-C in transition to fetal erythropoiesis. *Blood* **128**, 710–720 (2016).
5. Karpanen, T. *et al.* Lymphangiogenic growth factor responsiveness is modulated by postnatal lymphatic vessel maturation. *Am. J. Pathol.* **169**, 708–718 (2006).
6. Tabassum, D. P. & Polyak, K. Tumorigenesis: it takes a village. *Nat. Rev. Cancer* **15**, 473–483 (2015).
7. Karkkainen, M. J. *et al.* Vascular endothelial growth factor C is required for sprouting of the first lymphatic vessels from embryonic veins. *Nat. Immunol.* **5**, 74–80 (2004).
8. Mäkinen, T. *et al.* Inhibition of lymphangiogenesis with resulting lymphedema in transgenic mice expressing soluble VEGF receptor-3. *Nat. Med.* **7**, 199–205 (2001).
9. Nurmi, H. *et al.* VEGF-C is required for intestinal lymphatic vessel maintenance and lipid absorption. *EMBO Mol. Med.* (2015). doi:10.15252/emmm.201505731
10. Tammela, T. *et al.* VEGFR-3 controls tip to stalk conversion at vessel fusion sites by reinforcing Notch signalling. *Nat. Cell Biol.* **13**, 1202–1213 (2011).
11. Heinolainen, K. *et al.* VEGFR3 Modulates Vascular Permeability by Controlling VEGF/VEGFR2 Signaling. *Circ. Res.* **120**, 1414–1425 (2017).
12. Bazigou, E. *et al.* Genes regulating lymphangiogenesis control venous valve formation and maintenance in mice. *J. Clin. Invest.* **121**, 2984–2992 (2011).
13. Wang, Y. *et al.* Smooth muscle cell recruitment to lymphatic vessels requires PDGFB and impacts vessel size but not identity. *Dev. Camb. Engl.* **144**, 3590–3601 (2017).
14. Klotz, L. *et al.* Cardiac lymphatics are heterogeneous in origin and respond to injury. *Nature* **522**, 62–67 (2015).
15. Eilken, H. M. & Adams, R. H. Dynamics of endothelial cell behavior in sprouting angiogenesis. *Curr. Opin. Cell Biol.* **22**, 617–625 (2010).
16. Geudens, I. & Gerhardt, H. Coordinating cell behaviour during blood vessel formation. *Dev. Camb. Engl.* **138**, 4569–4583 (2011).
17. Niessen, K. *et al.* The Notch1-Dll4 signaling pathway regulates mouse postnatal lymphatic development. *Blood* **118**, 1989–1997 (2011).
18. Zheng, W. *et al.* Notch restricts lymphatic vessel sprouting induced by vascular endothelial growth factor. *Blood* **118**, 1154–1162 (2011).

REVIEWERS' COMMENTS:

Reviewer #1 (Remarks to the Author):

The authors have done a thorough and careful job in addressing all of my concerns. The new functional data, careful expression analysis and well balanced interpretation are convincing and improve the study. I congratulate the authors on a high quality, thorough and rigorous body of work.

Reviewer #2 (Remarks to the Author):

The authors provide a substantially revised manuscript which mostly addresses all essential critique points. Having said that, the question why the embryonic skin lymphatics do not show hyperproliferation of the non-targeted LECs similar to what the authors describe in the postnatal ear skin lymphatics has not been discussed. The new experiments and text added addresses whether VEGFR3 plays a role in cell survival and proliferation similarly in both embryonic and postnatal development of lymphatics, but seems to side-step the question why one situation leads to hypoplastic vessels and the other to hyperbranching networks. The manuscript would benefit from at least discussing this point, even if the true reason for these interesting differences cannot be resolved at this time.

In addition there are a few minor points that should be corrected.

1. The reference list shows duplication of a few references, for example Hellstrom et al., 2007. Please check carefully and correct.
2. The discussion page 16 line 341 and following. The sentence starting "Consistent with its role in blood vascular endothelial tip" is incorrect. Heterozygous or homozygous deletion of VEGFR3 does not impair the ability of cells to contribute to the tip position in the retina, unlike what the authors find for lymphatics. So the sentence should rather read "In contrast to .." or be omitted.
3. Finally, there are a few instances where words appear to be missing, a careful final proof read is recommended.

Reviewer #3 (Remarks to the Author):

I am fully satisfied with the answers, only couple of minor points:

1. Please include the data on the efficiency of Notch1 siRNA (Fig.7C)
2. "Although Notch inhibition promoted proliferation in LECs cultured in complete serum containing medium in the absence of VEGF-C, the effect was more pronounced when VEGF-C was present (Fig. 7c)" - please indicate whether this difference is statistically significant.
3. For the sake of clarity, I would suggest to keep only VEGFR3-IgG blocking data, MAZ51 and DC101 data do not add much of information.

NCOMMS-17-16093A

Response to the reviewers comments:

Reviewer #1 (Remarks to the Author):

The authors have done a thorough and careful job in addressing all of my concerns. The new functional data, careful expression analysis and well balanced interpretation are convincing and improve the study. I congratulate the authors on a high quality, thorough and rigorous body of work.

Reviewer #2 (Remarks to the Author):

The authors provide a substantially revised manuscript which mostly addresses all essential critique points. Having said that, the question why the embryonic skin lymphatics do not show hyperproliferation of the non-targeted LECs similar to what the authors describe in the postnatal ear skin lymphatics has not been discussed. The new experiments and text added addresses whether VEGFR3 plays a role in cell survival and proliferation similarly in both embryonic and postnatal development of lymphatics, but seems to side-step the question why one situation leads to hypoplastic vessels and the other to hyperbranching networks. The manuscript would benefit from at least discussing this point, even if the true reason for these interesting differences cannot be resolved at this time.

Response: The differential effect of *Vegfr3* deletion on embryonic and early postnatal vasculature is indeed interesting and many factors may be at play, including kinetics of vascular plexus formation and kinetics and efficiency of Cre-mediated recombination. We have not been able to resolve the reasons for the differences at this point, but we have added the following text in the discussion:

“Our results do not fully explain why *Vegfr3* deletion induces lymphatic vessel hyperplasia and hyperproliferation of non-targeted LECs in early postnatal skin but not in embryonic skin. Different kinetics of vascular plexus formation, but also the kinetics of gene deletion^{18,21} in the two situations (4-OHT – fast kinetics in embryos vs. Tamoxifen – slower kinetics in neonatal mice) may affect the outcome that is determined by a balance between cell-autonomous and non-cell-autonomous effects of *Vegfr3* deletion. In the embryo the primitive dermal vascular plexus forms within 2-3 days, but in the postnatal ear this process takes 10-12 days. It is possible that during the rapid expansion of embryonic vasculature the cell-autonomous effect of *Vegfr3* deletion on LEC proliferation and survival dominates. Another possibility is that a specific subset of LECs in the early postnatal vasculature are not efficiently targeted by the *Prox1-CreER*^{T2}, allowing for a sufficient number of LECs to escape recombination and respond to VEGFR3⁺ neighbors by proliferation. The observed efficient *Prox1-CreER*^{T2} driven recombination in wild type vasculature does not suggest that this is the case, however, such a scenario may become evident only when non-targeted cells show a selective growth advantage“.

In addition there are a few minor points that should be corrected.

1. The reference list shows duplication of a few references, for example Hellstrom et al., 2007. Please check carefully and correct.

Response: We have carefully checked the reference list and removed duplicates.

2. The discussion page 16 line 341 and following. The sentence starting “Consistent with its role in blood vascular endothelial tip” is incorrect. Heterozygous or homozygous deletion of VEGFR3 does not impair the ability of cells to contribute to the tip position in the retina, unlike what the authors find for lymphatics. So the sentence should rather read “In contrast to ..” or be omitted.

Response: We thank the Reviewer for noticing this mistake which has now been corrected. The corrected sentence is as follows: “LEC-specific deletion of *Vegfr3* in embryos led to dermal lymphatic vessel hypoplasia and revealed an indispensable function of VEGFR3 for lymphatic endothelial tip cell function and vessel sprouting.”

3. Finally, there are a few instances where words appear to be missing, a careful final proof read is recommended.

Response: We have carefully proof read the manuscript and hope that all mistakes have been corrected.

Reviewer #3 (Remarks to the Author):

I am fully satisfied with the answers, only couple of minor points:

1. Please include the data on the efficiency of Notch1 siRNA (Fig.7C)

Response: Western blot data showing the efficiency of NOTCH1 siRNA has been added as requested (Fig. 7c).

2."Although Notch inhibition promoted proliferation in LECs cultured in complete serum containing medium in the absence of VEGF-C, the effect was more pronounced when VEGF-C was present (Fig. 7c)" - please indicate whether this difference is statistically significant.

Response: The difference between VEGF-C untreated and treated groups show a significant difference with $P=0.0464$ (one-way ANOVA with Tukey's post hoc test). We have indicated this in the graph and the figure legend (now Fig. 7d). As per Nature Communications guidelines ('For small sample sizes ($n<5$) descriptive statistics are not appropriate, instead plot individual data points'), we have now also shown individual data points in Fig. 7d (as well as in Fig. 7b).

3. For the sake of clarity, I would suggest to keep only VEGFR3-IgG blocking data, MAZ51 and DC101 data do not add much of information.

Response: We believe that these data, and MAZ51 data in particular, are important as they provide evidence for anastomosis being a critical element of the hyperplasia phenotype. Specifically, the MAZ51 data show that excessive anastomosis is an active process driven by non-targeted VEGFR3⁺ cells when in contact with VEGFR3 deleted cells. These data provided an (at least partial) answer to

Reviewer 1's questions on the abnormal anastomosis process in the mutant vasculature. For these reasons we think that the data should be presented in the manuscript.